# Benchmarking Continuous Time Models for Predicting Multiple Sclerosis Progression

**Alexander Norcliffe**[*]                                     *alex.norcliffe98@gmail.com*
*University of Cambridge*

**Lev Proleev**                                                 *levp@google.com*
*Google Research*

**Diana Mincu**                                                 *dmincu@google.com*
*Google Research*

**Fletcher Lee Hartsell**                                       *fletcher.hartsell@duke.edu*
*Duke University Health System*

**Katherine Heller**                                           *kheller@google.com*
*Google Research*

**Subhrajit Roy**[†]                                           *subhrajitroy@google.com*
*Google Research*
**for MSOAC**[‡]

*Reviewed on OpenReview:* `https://openreview.net/forum?id=2uMnAwWnRy`

## Abstract

Multiple sclerosis is a disease that affects the brain and spinal cord, it can lead to severe disability and has no known cure. The majority of prior work in machine learning for multiple sclerosis has been centered around using Magnetic Resonance Imaging scans or laboratory tests; these modalities are both expensive to acquire and can be unreliable. In a recent paper it was shown that disease progression can be predicted effectively using performance outcome measures and demographic data. In our work we build on this to investigate the modeling side, using continuous time models to predict progression. We benchmark four continuous time models using a publicly available multiple sclerosis dataset. We find that the best continuous model is often able to outperform the best benchmarked discrete time model. We also carry out an extensive ablation to discover the sources of performance gains, we find that standardizing existing features leads to a larger performance increase than interpolating missing features.

An associated video presentation is available at: `https://www.youtube.com/watch?v=sqDLDkbP2H0`

## 1   Introduction

Multiple Sclerosis (MS) is the most common demyelinating disease (Leray et al., 2016) affecting approximately 2.8 million people worldwide (Lane et al., 2022), and being one of the leading causes of disability in

---

[*]Work done as a student researcher at Google Research from June-December 2022.

[†]Corresponding author.

[‡]Data used in the preparation of this article were obtained from the Multiple Sclerosis Outcome Assessments Consortium (MSOAC). As such, the investigators within MSOAC contributed to the design and implementation of the MSOAC Placebo database and/or provided placebo data, but did not participate in the analysis of the data or the writing of this report.

young adults (Coetzee & Thompson, 2020). It is a lifelong disease of the nervous system that can lead to a wide range of symptoms from fatigue to problems with movement and vision. Currently there is no known cure but the symptoms can be treated with medication and physical therapy, where the aim of a clinician is to limit progression. Therefore, an effective application of machine learning methods in this space would be to predict progression of MS as well as informing interventions and treatment options to obtain the best prognosis for a given patient.

Prior machine learning works that investigate MS progression typically rely on Magnetic Resonance Imaging (MRI) scans or laboratory tests (Pinto et al., 2020; Zhao et al., 2017; Seccia et al., 2020; Tommasin et al., 2021). These modalities are both expensive to obtain (MRI scans cost on the order of thousands of US dollars) and infrequent, added to this they are not always reliable (Whitaker et al., 1995). The progression of MS has also been studied using clinical data (Lawton et al., 2015; Tozlu et al., 2019; De Brouwer et al., 2021) with models ranging from Latent Class Linear Mixed Models, to time-aware GRU.

To this end, in a previous study Roy et al. (2022) demonstrated that it is possible to use both performance outcome measures and demographic data to predict disease progression. Performance outcome measures (POMs) are a time-series that contain either responses to questionnaires or results from physical tests to determine MS progression (Rudick et al., 2002). These tests are designed to evaluate the functions affected by MS, and include walking, balance, cognition and dexterity tests. Roy et al. (2022) showed that POMs and demographic data successfully predict progression for a range of models, from classical models such as logistic regression to deep learning models for sequences such as Temporal Convolutional Network (TCN) (Bai et al., 2018). However, these models are not suited to irregular time-series with missing features, and various techniques such as imputing with zeros and binning are applied to make the data compatible with the models, which can add bias. In contrast, these realistic hurdles of irregular and missing measurements can be handled in principle by continuous time models. Continuous time models such as Neural ODEs (Chen et al., 2018) learn to model an instantaneous velocity of a state to calculate the time evolution. They have three key advantages: (1) they naturally handle irregular time-series; (2) via the use of control signals missing features can be interpolated and therefore are imputed with more physically plausible values (Kidger et al., 2020); (3) continuous time models enforce the inductive bias that the data is generated from a dynamical process such as biological interactions (Alharbi & Rambely, 2020). Neural Differential Equations have been applied in the medical setting (Kidger et al., 2020; Morrill et al., 2021; Qian et al., 2021; Seedat et al., 2022; Gwak et al., 2020), however to our best knowledge we are the first to apply these to MS prediction using POMs. Our *primary* contributions are: (1) We benchmark four continuous time models on a public MS dataset, and see that the best continuous model outperforms the previously best benchmarked discrete model. (2) We test the use of interpolation to fill missing values rather than fixed values, in this setting we find the difference is negligible. To carry this out we faced various challenges requiring technical solutions: (1) We adapt the framework of Roy et al. (2022) to handle continuous models which impute by interpolation. (2) We generalize an existing method by Morrill et al. (2021) to train continuous models with irregular data in batches. (3) We introduce two new interpolation schemes using monotonically increasing cubics to interpolate missing values and be control signals for models. We view these as *secondary* contributions and emphasize that the main message of the paper is that continuous models can be applied successfully in the MS setting.

## 2 Background

### 2.1 Neural Differential Equations

Here we briefly introduce the concepts behind continuous time models. For more information we encourage the reader to see resources such as Chen et al. (2018); Dupont et al. (2019); Massaroli et al. (2020); Kidger et al. (2020); Li et al. (2020); Kidger (2022). For a hidden state $\boldsymbol{h} \in \mathbb{R}^n$, and time $t \in \mathbb{R}$, Neural Ordinary Differential Equations (Neural ODEs) (Chen et al., 2018) model the instantaneous rate of change of $\boldsymbol{h}$ with respect to $t$ using a neural network $f : \mathbb{R}^n \times \mathbb{R} \times \Theta \to \mathbb{R}^n$, with parameters $\theta \in \Theta$

$$\frac{d\boldsymbol{h}}{dt} = f(\boldsymbol{h}, t, \theta). \tag{1}$$

For brevity we now use the convention that $f(\boldsymbol{h}, t, \theta)$ is written as $f_\theta(\boldsymbol{h}, t)$. Given an initial condition $\boldsymbol{h}(t_0) = \boldsymbol{h}_0$, we can find the solution at general time $\boldsymbol{h}(t)$ using ODE solvers. The simplest is the Euler solver, where we start from the known $\boldsymbol{h}(t_0)$ and repeatedly apply the update rule $\boldsymbol{h}(t + \Delta t) = \boldsymbol{h}(t) + f(\boldsymbol{h}(t), t)\Delta t$, this approaches the true solution as $\Delta t$ approaches 0. For more accurate solutions we can look to the existing numerics work and use advanced black box ODE solvers, for example Dormand-Prince (Dormand & Prince, 1980), to approximately solve the initial value problem and calculate the hidden state at a later time

$$\boldsymbol{h}(t) = \mathrm{ODESolve}(f_\theta, \boldsymbol{h}_0, t_0, t). \tag{2}$$

Crucially, we can differentiate the solutions of the ODE solve with respect to the parameters $\theta$, the initial state $\boldsymbol{h}_0$, initial time $t_0$ and the solution times $t$. This can be done by using the adjoint sensitivity method (Chen et al., 2018; Pontryagin, 1987) or by differentiating through the solver operations, since solver operations themselves are differentiable. This allows us to backpropagate through the ODE solve and train any learnable parameters via gradient descent. Therefore, Neural ODEs can be used as layers in a network, for example, as a feedforward layer where we solve the ODE from $t_0$ to a terminal time $t_1$; or they can be used to output a series by calculating the hidden state at an ordered list of times $\{t_0, t_1, ..., t_n\}$.

One advantage of Neural ODEs is that they intrinsically deal with irregular time-series, without the need for bucketing. This is because the model is able to deal with continuous time by design by modeling the instantaneous velocity, rather than a discontinuous update. Moreover, they use the inductive bias that a time-series is generated from an underlying dynamical process, that is, the instantaneous rate of change of the current state depends on the state itself. For example, in epidemiology, compartmental models (Tang et al., 2020) are used to determine how a disease spreads through a population; or in virology the concentration of virus, antibodies and infected cells can be modelled with differential equations (Perelson et al., 1996). In our case, the progression of MS is driven by biological mechanisms so MS progression can also be modeled with differential equations. We do not know what these differential equations are, and therefore Neural ODEs are well suited to the problem by combining the inductive bias of an underlying dynamical system with the universal approximation power of neural networks (Cybenko, 1989).

## 2.2 Interpolation and Control Signals

In many real world cases data is incomplete, to overcome this, missing values are imputed. Often this is done by filling with zero or the mean value of a feature. In time-series applications we are able to use already observed values to plausibly fill in missing values based on the rest of the series. For example we may have a time-series $((x_0, t_0), (\emptyset, t_1), (x_2, t_2))$, we can linearly interpolate, giving $((x_0, t_0), (\tilde{x}_1, t_1), (x_2, t_2))$, where $\tilde{x}_1 = \frac{x_2 - x_0}{t_2 - t_0}(t_1 - t_0) + x_0$. More generally, we can calculate $X(t)$, which is an interpolation of the observed time-series (Davis, 1975; Lunardi, 2009). The values of $X$ match the observations exactly, $X(t_n) = x_n$, and will also predict a value for unobserved features. $X(t)$ can be evaluated at *all* times not just observation times, giving us a *control signal* based on the observations.

## 3 Continuous Modeling Framework

In this work we consider continuous models. The models are adapted for this setting, via our new framework building on that of Roy et al. (2022). We recommend seeing Figure 1 and Section 3 of that work for information on this stage of the pipeline. Briefly, the preprocessing consists of converting raw data to a common representation specifically for discrete models, splits subjects into train/test sets, fills missing values with zeros and pads the time-series for batching; the postprocessing is able to calculate metrics for the whole population as well as subgroups. To adapt to the continuous setting, the models receive an interpolation of the dynamic features $X$ (the POMs at different timestamps) known as the control signal and a vector of the static features $\boldsymbol{c}$ known as the context (such as sex and ethnicity). They then make a prediction at the given evaluation times, see more about the specific models and their architectures in Section 4.

**Passing an interpolation to a model.** Since a control signal is continuous, we cannot pass the whole interpolation to a model, as there are infinitely many points. This is instead achieved by making the control

signals piece-wise cubic functions. So that between $t_i$ and $t_{i+1}$, $X(t) = a_i(t - t_i)^3 + b_i(t - t_i)^2 + c_i(t - t_i) + d_i$, then the coefficients $(a_i, b_i, c_i, d_i)$ are passed to the model to compactly represent the interpolation.

**Batching using Pseudo-Time.** A common requirement of Deep Learning training is being able to train using batches, this is non-trivial for irregular time-series. The original high level solution (Chen et al., 2018) for continuous models is to merge all of the times in the different samples (concatenate and sort). The ODE is then solved for *all* of the merged times, then a binary mask is applied to extract the relevant predictions for each time-series. Since then, low-level implementations of parallel ODE solvers have been introduced (Lienen & Günnemann, 2022; Kidger, 2021) to the PyTorch (Paszke et al., 2019) and Jax libraries (Bradbury et al., 2018). These use a separate ODE solver for each series in the batch, all being run in parallel. This requires specialist parallel hardware - GPUs, which is typically not the most restrictive requirement. Our models and experiments were implemented in TensorFlow and Keras (Abadi et al., 2015; Chollet et al., 2015), preventing us using these low level solutions. Instead of implementing this and diverging from the main aim of the paper, we opt for a different high level solution. This is a generalization of the method given in Morrill et al. (2021) from Online Neural CDEs to all continuous methods. We use the integers as a regularly spaced pseudo-time, $s$. And instead of interpolating the time-series with the true timestamps $t$, we interpolate with $s$ giving $X(s)$. We also interpolate the true time using $s$, so that $t = \psi(s)$. Then, if the dynamics function we learn represents the rate of change of a state with respect to *true time*

$$\frac{d\boldsymbol{h}}{dt} = f_\theta(\boldsymbol{h}, t).$$

By adapting this to the $s$ domain such that the rate of change with respect to *pseudo-time* is

$$\frac{d\boldsymbol{h}}{ds} = f_\theta(\boldsymbol{h}, \psi(s))\frac{d\psi}{ds}$$

we can show that $\text{ODESolve}(\frac{dh}{dt}, h_0, t_0, t) = \text{ODESolve}(\frac{dh}{ds}, h_0, s_0, s)$. Note this is only true if $\psi$ is an increasing, surjective and piece-wise differentiable function (which can be achieved with certain interpolation schemes). This allows us to straightforwardly batch as we can solve all of the ODEs in the batch in the same regular $s$ domain, and shift each to their respective true time domains. We use this strategy over merging the times since the interpolations are already calculated for the models to run, thus we are simply re-purposing them to achieve the same result. As well as this, we do not need to solve the ODE for all *merged true times*, only the *shared regular pseudo-times*, we do not store as many intermediate ODE solution values. This suggests that this method *could* be more memory efficient than the merging method and compute efficient for certain models, however, this was not explicitly tested. See Appendix D for a full explanation of the theoretical gains as well as proof of equality & requirements of $\psi$. As well as using pseudo-time, we must pad sequences to the same length, keeping track of which observations are real and which are imputed in order to train in batches. To achieve this, an initial continuous model specific data preprocessing step is used.

## 3.1 Data Preprocessing

A subject from the data consists of a vector of context features $\boldsymbol{c}$, and timestamped time-series of sequence features $\boldsymbol{x}(t)$, label $y(t)$ & sample weights $w(t)$. The sample weights consist of 1s and 0s and inform the loss function and evaluation metric if a prediction is used in the calculation, if $w(t) = 1$ we care about how close $\hat{y}(t)$ is to $y(t)$ (we can set $w(t) = 0$ when we have a missing label). The sequences are irregular with varying lengths, both the sequences and contexts may have missing values. We wish to convert this so that all time-series are the same length, with missing features imputed and the irregular times dealt with so that we can train efficiently using minibatches. Therefore the preprocessing consists of these steps in order:

1. Standardize the sequences and the context feature-wise using only the observed values so they have mean 0 and standard deviation 1.
2. Impute missing *context* features with a constant. We choose zero since that is now the mean.
3. Pad sequences to the same length. We fill: sequence features with missing values (nans); observation times with the last observed time; labels with zero; sample weights with zero.
4. Calculate a cumulative sum of the number of observations in each sequence feature-wise, such that if there is a missing value this count does not increase. This will be called the *observation count*,

denoted $\boldsymbol{o}$. When we impute later this provides information to the model that either we have a true observation or an imputed one, as required by (Kidger et al., 2020).

5. Concatenate $(\boldsymbol{x}, \boldsymbol{o}, t)$ along the feature dimension, so that these are the new sequence features.
6. Convert this to a physically plausible dataset, where future observations do not affect the past (see Section 3.2 for explanation and two methods to do this).
7. Interpolate the new physically plausible time-series using integers as the pseudo-time $s$, in accordance with our new batching strategy. Giving piece-wise cubic coefficients.

This produces a series of coefficients that represent piece-wise cubics as the interpolating function for each feature. It is computationally expensive to calculate the coefficients each time they are needed, so they are calculated *once* and stored. They replace $(\boldsymbol{x}, \boldsymbol{o}, t)$ in the dataset as input to the model, since the sequence is easy to recover using the coefficients. After this preprocessing each subject in the data has a tuple of: $((\text{coeffs}, \boldsymbol{c}), y, w)$. Note that $y$ and $w$ are also longitudinal just like coeffs as we make predictions at every timestep. This preprocessing means every time-series in the dataset is the same length (due to padding) and missing values are filled (either with the mean or interpolated values). After preprocessing we can shuffle the dataset along the first dimension giving us minibatches, using pseudo-times allows us to calculate the predictions of a batch even for irregular times, and the sample weights account for padding and missing labels in the loss calculation. Therefore, we can now train a continuous model by repeatedly giving it a minibatch, calculating the loss and using an optimizer of choice. We can also calculate the performance metrics in the same batched way.

### 3.2 Enforcing Physical Solutions

It is critical that a model's predictions are physically plausible, i.e. the model only uses the current and previous measurements to make a prediction. At inference time we do not have access to future measurements, so the model must be trained such that predictions at $t$ only use the measurements up to $t$. Control signals in general are not physical, they either use the whole time-series to interpolate, or use the next observed value which is not physical when there are missing values (i.e. we skip a missing value and use a future observed value). To overcome this we enforce physical solutions in continuous models in two ways:

1. Reprocess the dataset, by making many copies of a series (see Reprocessing the Dataset below).
2. Use continuously online control signals, called "recti" control signals (see Continuously Online Interpolation below).

**Reprocessing the Dataset.** For each subject in a series we make copies of the subject so that for the $i$-th timestamp, we consider the time-series up until that point, and fill forward the $i$-th observation until the end. Then we set the sample weights to 0 everywhere except at point $i$ where it is 1. This means that only previous observations are used to predict at timestamp $i$, and then by adjusting the sample weights we only consider the $i$-th label for that copy in the loss. We cannot consider earlier because then the model is using observations up until $i$ to make predictions for points earlier than $i$, we don't predict later because it is not relevant to do so. For example, this time-series is transformed below:

$$
\left[\begin{array}{ccc} \left[\begin{array}{c} x_1 \\ t_1 \\ y_1 \\ 1.0 \end{array}\right] & \left[\begin{array}{c} x_2 \\ t_2 \\ y_2 \\ 1.0 \end{array}\right] & \left[\begin{array}{c} x_3 \\ t_3 \\ y_3 \\ 1.0 \end{array}\right] \end{array}\right]
\longrightarrow
\left[\begin{array}{c} \left[\begin{array}{ccc} \left[\begin{array}{c} x_1 \\ t_1 \\ y_1 \\ 1.0 \end{array}\right] & \left[\begin{array}{c} x_1 \\ t_1 \\ y_1 \\ 0.0 \end{array}\right] & \left[\begin{array}{c} x_1 \\ t_1 \\ y_1 \\ 0.0 \end{array}\right] \end{array}\right] \quad \left[\begin{array}{ccc} \left[\begin{array}{c} x_1 \\ t_1 \\ y_1 \\ 0.0 \end{array}\right] & \left[\begin{array}{c} x_2 \\ t_2 \\ y_2 \\ 1.0 \end{array}\right] & \left[\begin{array}{c} x_2 \\ t_2 \\ y_2 \\ 0.0 \end{array}\right] \end{array}\right] \\ \left[\begin{array}{ccc} \left[\begin{array}{c} x_1 \\ t_1 \\ y_1 \\ 0.0 \end{array}\right] & \left[\begin{array}{c} x_2 \\ t_2 \\ y_2 \\ 0.0 \end{array}\right] & \left[\begin{array}{c} x_3 \\ t_3 \\ y_3 \\ 1.0 \end{array}\right] \end{array}\right] \end{array}\right]
$$

Note that in the interest of space on the page we have wrapped the repeated subjects in the transformed time-series. We have highlighted in red where the copies are created, so the time series initially has shape $\mathbf{1} \times 3 \times 4$ and the resulting series has shape $\mathbf{3} \times 3 \times 4$, so we are extending along the batch dimension. This is

physical for all models because the actual data itself only contains previous observations, and is then the last measurement repeated. The downside is that this can make the dataset increase size significantly without introducing any new information.

**Continuously Online Interpolation (recti method).** To avoid our dataset growing exponentially we can also use continuously online interpolation schemes also called the recti method (Morrill et al., 2021). These extend the length of the time-series but not the number of samples. Let $\tilde{x}$ be the fill-forward of $x$, so any missing observations are replaced by the most recent observed value. For a time-series of length $n$ we then construct a "recti" time-series of length $2n - 1$. Where $X(2i - 1) = (\tilde{x}_i, t_i, y_i, w_i)$ for $i \in \{1, ..., n\}$ and $X(2i) = (\tilde{x}_i, t_{i+1}, y_i, 0.0)$ for $i \in \{1, ..., n - 1\}$. For example, this time-series is transformed below:

$$
\left[ \left[ \begin{bmatrix} x_1 \\ t_1 \\ y_1 \\ 1.0 \end{bmatrix} \begin{bmatrix} x_2 \\ t_2 \\ y_2 \\ 1.0 \end{bmatrix} \begin{bmatrix} x_3 \\ t_3 \\ y_3 \\ 1.0 \end{bmatrix} \right] \right] \longrightarrow \left[ \left[ \begin{bmatrix} \tilde{x}_1 \\ t_1 \\ y_1 \\ 1.0 \end{bmatrix} \begin{bmatrix} \tilde{x}_1 \\ t_2 \\ y_1 \\ 0.0 \end{bmatrix} \begin{bmatrix} \tilde{x}_2 \\ t_2 \\ y_2 \\ 1.0 \end{bmatrix} \begin{bmatrix} \tilde{x}_2 \\ t_3 \\ y_2 \\ 0.0 \end{bmatrix} \begin{bmatrix} \tilde{x}_3 \\ t_3 \\ y_3 \\ 1.0 \end{bmatrix} \right] \right]
$$

As before we have highlighted in red where the repeats occur. The original shape of this series was $1 \times \mathbf{3} \times 4$ and becomes $1 \times \mathbf{5} \times 4$ so we are extending in the time dimension rather than number of subjects. Crucially with the data rectified, we must then use what is known as a discretely online interpolation scheme. These only use the next observation to interpolate and since the observations are repeated[1] this remains physically plausible. Examples include Linear interpolation and Hermite Cubic interpolation (see Appendix F). With this, the model is similarly only able to use previous observations to make a prediction. This method is only valid for models which already enforce physical solutions by design, such as TCN. However, later observations are still included in the time-series, so models that are not physical by design (i.e. those that use a whole time-series to predict at each point) require the method of reprocessing the data.

### 3.3 Interpolation Schemes

Here we give the interpolation schemes that are used in the continuous models. We give short overviews of each one here, for full information including equations, and advantages & disadvantages of each one see Appendix F. The existing interpolation schemes used are described in full in Morrill et al. (2021). They are:

- **Linear:** This linearly interpolates the points with a straight line between observations, the $a$ and $b$ coefficients are zero. It can be used as an increasing, surjective, piece-wise differentiable interpolation so can be used to interpolate the time-channel for batching with pseudo-time.
- **Hermite Cubic:** This takes the linear interpolation, and now the previously unused $a$ and $b$ coefficients are used to smooth the interpolation. The first derivative is continuous at observations by enforcing the derivative to be equal to the gradient between that observation and the previous. This ODE is faster to solve than with linear interpolation due to having a continuous derivative.
- **Natural Cubic:** This is a cubic interpolation that enforces continuity in the interpolation, the derivative and the second derivative. This makes the interpolation the most "natural" for the observed points, however, the whole time-series is used to interpolate so is not physically plausible.
- **Rectilinear:** This interpolation uses the recti method of extending along the time dimension described previously, and then linearly interpolates those points. It is a continuously online control signal. However, since it is linear it can still be slow to solve the ODE.

We also introduce and investigate two novel interpolation schemes:

- **Monotonic Cubic:** This takes the Hermite Cubic described previously, but rather than enforcing a derivative that would be obtained by linearly interpolating backwards, we enforce zero derivative at observation points. The purpose is that it is possible to interpolate increasing measurements with an increasing, surjective, piece-wise differentiable function. This allows us to interpolate the time channel to guarantee batching with pseudo-time.
- **Recticubic:** This uses the recti system of extending the length of the time-series. In contrast to the rectilinear interpolation, we then use Hermite cubics to interpolate the features, and monotonic cu-

---

[1]Repeating the label does not introduce any bias since the sample weights are zero for repeated labels.

bics to interpolate the observation and time channels. This constructs a smooth physically plausible interpolation that also respects the requirements for interpolation of the time channel.

Note, since we are using the pseudo-time method to train in batches, we still require an increasing, surjective, piece-wise differentiable interpolation of $t = \psi(s)$. So for the linear and rectilinear schemes we use linear interpolations of the features, observations and $t$. And for all other control signals we use the monotonic cubic interpolation for the observations and $t$ and the named scheme for interpolating the features.

## 4 Models

In this work we consider four neural differential equation based models. They are adapted for this setting, we recommend seeing their original papers for a full description, here we describe them as implemented, these are also shown pictorially in Figure 1 which we also recommend following alongside the descriptions. All models receive (coeffs, $c$) as input, where $c$ is the context and the coeffs can be used to create a control signal of the time-series, $X(s)$. For notational purposes, a $c$ superscript on a function denotes the function takes the context as additional input, $f_\theta^c(h, t) \equiv f(h, t, c, \theta)$. This is so we can keep parameters and context nominally separate from the hidden state and time, which dynamically change during the ODE solve.

### 4.1 Neural Controlled Differential Equations

The dynamics of a Neural CDE (Kidger et al., 2020) uses the whole control signal (not just $\psi(s)$) to constantly include information from the time-series in the dynamics.

$$\frac{dh}{ds} = f_\theta^c(h, \psi(s))\frac{dX}{ds}$$

Where $f_\theta^c$ outputs a matrix since $X$ and its derivatives are vectors. Note that since $\psi$ is the last dimension of $X$ we implicitly include its derivative here for batching purposes. The model embeds the first observation to an initial condition in a latent space with $e_\phi^c$, solves the above dynamics in the latent space and predicts the labels from the hidden state at a given time with $d_\omega$.

$$h_0 = e_\phi^c(X(0))$$

$$h_i = \text{ODESolve}(\frac{dh}{ds}, h_0, 0, i)$$

$$\hat{y}_i = d_\omega(h_i)$$

### 4.2 ODE-RNN

The ODE-RNN model (Rubanova et al., 2019) generalizes RNN models so the hidden state evolves according to a differential equation *between* observations.

$$\frac{dh}{ds} = f_\theta^c(h, \psi(s))\frac{d\psi}{ds}$$

And then *at* observations the hidden state undergoes a discontinuous change. See the top right of Figure 1 for a diagram of this, as well as the equations of the method below. The initial hidden state is a vector of zeros. This evolves continuously between observations and at observations changes discontinuously based on its current value, the observation and context using $g_\phi^c$. Finally predictions are made from the hidden state at a given time with $d_\omega$.

$$h_{-1} = \mathbf{0}$$

$$\tilde{h}_i = \text{ODESolve}(\frac{dh}{ds}, h_{i-1}, i-1, i)$$

$$h_i = g_\phi^c(\tilde{h}_i, X(i))$$

$$\hat{y}_i = d_\omega(h_i)$$

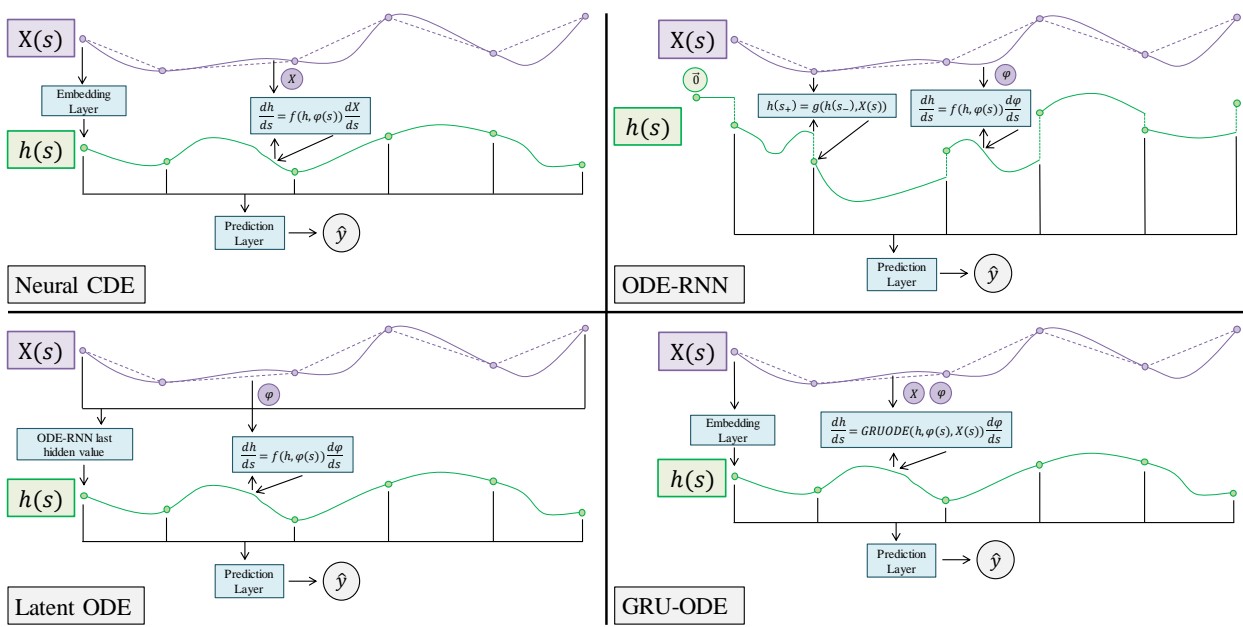

Figure 1: Block diagram of the four continuous models. The interpolation of $X(s)$ shows a linear (dashed) and a cubic (solid) interpolation. Diagrams are best viewed zoomed in.

The interpolation rules mean that $d\psi/ds$ is zero outside the range of the time-series, so the first update from $i = -1$ to $i = 0$ is only a discontinuous one, since the ODE function is zero here. A technical detail is that for both the ODE-RNN and Latent ODE below, during preprocessing we create an update mask for each subject at each timestep $u_i \in \{0, 1\}$, where 1 indicates we want to update the hidden state. The discontinuous update is instead $\boldsymbol{h}_i = u_i g_\phi^c(\tilde{\boldsymbol{h}}_i, X(i)) + (1 - u_i)\boldsymbol{h}_{i-1}$. So for padded data the update is zero.

### 4.3 Latent ODE

The Latent ODE extends the ODE-RNN model to an encoder-decoder model that encodes entire sequences and then decodes them. We use an ODE-RNN model to encode the whole sequence to a single vector, taking the final value of the hidden state from the ODE-RNN. This is then fed into another ODE to decode, where the decoding ODE is given by

$$\frac{d\boldsymbol{h}}{ds} = f_\theta^c(\boldsymbol{h}, \psi(s))\frac{d\psi}{ds}$$

The outputs of the decoding ODE are then fed through a final prediction layer $d_\omega$.

$$\boldsymbol{h}_0 = \text{ODERNN}_\phi^c(X)[\text{final hidden value}]$$

$$\boldsymbol{h}_i = \text{ODESolve}(\frac{d\boldsymbol{h}}{ds}, \boldsymbol{h}_0, 0, i)$$

$$\hat{y}_i = d_\omega(\boldsymbol{h}_i)$$

When encoding the sequence with the ODE-RNN, we encode backwards such that the earliest observation has the largest effect on the initial condition of the decoding ODE. Using update masks in the backwards ODE-RNN is particularly crucial here so that the padded values do not affect the encoding of the hidden state. Note that the Latent ODE encodes entire sequences, so this does not enforce physical solutions and we cannot use rectilinear or recticubic control signals.

### 4.4 GRU-ODE

The GRU-ODE (De Brouwer et al., 2019) is a continuous analogue of the Gated Recurrent Unit (Cho et al., 2014). The dynamics are given by

$$r = \sigma\big(W_{rx}X(s) + W_{rh}\boldsymbol{h} + W_{rc}\boldsymbol{c} + b_r\big)$$
$$z = \sigma\big(W_{zx}X(s) + W_{zh}\boldsymbol{h} + W_{zc}\boldsymbol{c} + b_z\big)$$
$$g = \tanh\big(W_{gx}X(s) + W_{gh}(r \odot \boldsymbol{h}) + W_{gc}\boldsymbol{c} + b_g\big)$$
$$\frac{d\boldsymbol{h}}{ds} = \big(1 - z\big) \odot \big(g - \boldsymbol{h}\big)\frac{d\psi}{ds}$$

for given weight matrices and bias vectors. These dynamics also depend on time since $t = \psi(s)$ is the last dimension of $X(s)$. We then follow the same steps as the Neural CDE, embed with $e_\phi^c$, solve the dynamics in the latent space and predict with $d_\omega$.

$$\boldsymbol{h}_0 = e_\phi^c(X(0))$$
$$\boldsymbol{h}_i = \text{ODESolve}(\frac{d\boldsymbol{h}}{ds}, \boldsymbol{h}_0, 0, i)$$
$$\hat{y}_i = d_\omega(\boldsymbol{h}_i)$$

All embedding and dynamics functions are given by multilayer perceptrons with ReLU hidden activations.[2] Embedding layers use one hidden layer and dynamics functions use two. The embedding layers use no final non-linearity, the dynamics functions use tanh to improve stability of the hidden dynamics. All prediction layers are single linear layers which end in non-linearities specific to the problem: sigmoid for binary classification, softmax for multiclass classification and no non-linearity for regression.

## 5 Experiments

Our experiments are designed to investigate how well the continuous baseline methods can perform in the same MS setting as non-continuous methods. Therefore, we test the models on the same dataset from Roy et al. (2022) and compare against the best performing model from that paper for a given task.

**Hyperparameters.** We found hyperparameters using a grid search and 10 fold cross-validation. The final hyperparameter configurations and values tested are given in Appendix B. Using cross-validation gives both different parameter initializations and train data splits to obtain uncertainty estimates on performance. Models are trained for 50 epochs with the Adam optimizer (Kingma & Ba, 2015), learning rates and batchsizes are hyperparameters given in Appendix B.

**Dataset Description.** The Multiple Sclerosis Outcome Assessments Consortium (MSOAC) (Rudick et al., 2014) is a partnership whose goal is to collect, standardize, and analyze data about MS. One such dataset which we use is the Placebo dataset, which consists of 2465 separate patient records from 9 clinical trials. Since this is a public dataset, Institutional Review Board (IRB) approval is not required. The features contain demographics, medical history, MS specific data and POMs. The labels are constructed from Expanded Disability Status Scale (EDSS) scores (Kurtzke, 1983). These are clinically annotated scores to determine the severity of the MS, the scores range from 0 to 10 in steps of 0.5. EDSS scores are partitioned into four distinct subsets: 0 - 1 for no disability, 1.5 - 2.5 for mild disability, 3 - 4.5 for moderate disability, and 5 - 10 for severe disability. As in Roy et al. (2022) the aim is to predict the EDSS score for a patient for four given prediction horizons. Due to the irregularity of clinical visits these are split into prediction windows: 0 - 6 months; 6 - 12 months; 12 - 18 months and 18 - 24 months, over the window the EDSS scores taken in clinic are averaged to $EDSS_{mean}$, if labels are still missing we set the sample weight $w$ to 0 so the prediction is not used to calculate metrics or the loss. We look at four subtasks, for each one we are trying to answer a question about the EDSS score in the future prediction window for each subject: predicting the EDSS score

---

[2]Except for GRU-ODE where the dynamics are restricted as described.

directly as a regression problem; predicting if EDSS > 3 as a binary classification problem; predicting if EDSS > 5 as a binary classification problem and predicting EDSS severity category as a multiclass classification problem. These final three are clinically insightful tasks, since they indicate the category of disability as well as if there has been a significant change in the MS in a given window. When predicting EDSS directly we report root mean squared error (RMSE) and train with mean squared error. For the binary classification tasks, due to class imbalance, we report the area under the precision recall curve (AUPRC) and train with binary cross entropy. When predicting the EDSS as a severity category we calculate the AUPRC for each class in a one vs rest approach and report the mean across classes and train with the cross entropy.

Rather than using all features we use a reduced set of PASAT, SDMT, NHPT, T25FW, EDSS, Age and Sex. These correspond to Paced Auditory Serial Addition Test, Symbol Digit Modalities Test, Nine Hole Peg Test, Timed 25-Foot Walk and the current EDSS score. These are the majority of the functional tests, removing most demographics and all questionnaires from the full feature set. We use a reduced set because interpolation and training are computationally expensive, the cost of the interpolation is $\mathcal{O}(n_f n_s n_t)$ where $n_f$ is the number of features, $n_s$ is the number of subjects and $n_t$ is the length of the series. Additionally, for every call of the Neural CDE dynamics function there is a matrix-vector multiplication of cost $\mathcal{O}(n_f)$. We also initially found continuous models were unstable to many features, we carry out a feature ablation in Appendix C.2 where we confirm this reduced set performs best. We specifically use these features because they have the lowest attrition compared to other features, they are approximately 10 times more abundant at the start of the study increasing to roughly 1000 times more abundant at 9 months (see Figure 2 of Roy et al. (2022) for a chart tracking the feature sparsity over time).

### 5.1 Results and Analysis

**Overall Performance.** We compare against the previously best performing model TCN, the performance has been attributed to the model's awareness of temporal evolution, in particular for predicting long term time-series. The results are given in Table 1. We see that in ten out of sixteen cases the Neural CDE outperforms TCN, often quite substantially. In some cases the Neural CDE performs worse than TCN, in particular for predicting the EDSS severity category it is consistently worse. We hypothesize this is due to the features used. Whilst Neural CDE performs worse with all features, it is likely that a different subset is optimal for this task. We also see that the other continuous models are occasionally able to beat TCN, but this isn't consistent. This shows that being continuous does not guarantee performance gains, but it also does not lead to performance decrease (Neural CDE was able to beat TCN, and the other three methods typically beat all other methods from Roy et al. (2022)). The insight gained is that the correct model with the right hyperparameters must still be used for the problem, we have also further supported the use of Neural CDE over other continuous models. Neural CDE outperforming ODE-RNN, Latent ODE and GRU-ODE is consistent with the findings in (Kidger et al., 2020), this is likely due to Neural CDEs having provably more representation power acting on sequences than the other continuous models (see Kidger et al. (2020) for these theorems). To further support the use of Neural CDE we also newly benchmark RNN in Appendix A where we see that RNN performs worse than TCN and Neural CDE.

**Standardization and Interpolation Ablation.** Here we investigate how the performance changes based on whether we standardize features or interpolate missing values. When we don't impute with interpolated values we do so with zero instead. Note that interpolation must still be done to construct control signals $X(s)$, this ablation is about how we impute the missing values. Results for the Neural CDE are given in Table 2. We see that for Neural CDEs standardizing the values has the largest positive effect on performance. This is likely due to standardization transforming $X$ into a manageable range, and by extension $dX/ds$, so that the dynamics of the Neural CDE are well regularized. We run the same ablations on the other models in Appendix C, Tables 11-13, they support this hypothesis since they have far less significance between standardizing and not standardizing features and their dynamics do not end in the matrix vector multiplication $dX/ds$, only the vector scalar multiplication $d\psi/ds$. We see that interpolating or filling with zeros does not have a significant effect; when standardization is fixed the difference between interpolating and filling with zero is almost negligible. We carry out the same ablation of the other continuous models in Appendix C and see similar results, therefore standardization is crucial for performance but interpolating missing values is not.

Table 1: MSOAC Values against the best performing model from (Roy et al., 2022). All metrics are AUPRC, except for the $EDSS_{mean}$ task, where we use RMSE. Best values are in **bold** and values better than TCN but not best are underlined.

| Prediction Task | Prediction Window | TCN | Neural CDE | ODE RNN | Latent ODE | GRU ODE |
|---|---|---|---|---|---|---|
| $EDSS_{mean}$ | 0 - 6 mo | **1.264 ± 0.055** | 1.408 ± 0.059 | 1.768 ± 0.139 | 1.617 ± 0.072 | 2.312 ± 0.271 |
| | 6 - 12 mo | 1.650 ± 0.067 | **1.627 ± 0.055** | 1.886 ± 0.248 | 1.776 ± 0.071 | 2.236 ± 0.207 |
| | 12 - 18 mo | 1.725 ± 0.074 | **1.652 ± 0.066** | 1.826 ± 0.123 | 1.810 ± 0.090 | 2.066 ± 0.086 |
| | 18 - 24 mo | 1.666 ± 0.128 | **1.587 ± 0.078** | 1.741 ± 0.100 | 1.707 ± 0.097 | 1.834 ± 0.107 |
| $EDSS_{mean}$ > 3 | 0 - 6 mo | **0.909 ± 0.014** | 0.908 ± 0.010 | 0.869 ± 0.018 | 0.871 ± 0.017 | 0.865 ± 0.021 |
| | 6 - 12 mo | 0.820 ± 0.027 | **0.852 ± 0.016** | 0.823 ± 0.024 | 0.819 ± 0.030 | 0.788 ± 0.025 |
| | 12 - 18 mo | 0.768 ± 0.031 | **0.797 ± 0.027** | 0.766 ± 0.032 | 0.776 ± 0.027 | 0.730 ± 0.041 |
| | 18 - 24 mo | 0.703 ± 0.038 | **0.742 ± 0.035** | 0.697 ± 0.044 | 0.703 ± 0.044 | 0.666 ± 0.062 |
| $EDSS_{mean}$ > 5 | 0 - 6 mo | 0.848 ± 0.035 | **0.872 ± 0.021** | 0.807 ± 0.032 | 0.788 ± 0.026 | 0.794 ± 0.036 |
| | 6 - 12 mo | 0.722 ± 0.039 | **0.793 ± 0.039** | 0.751 ± 0.026 | 0.740 ± 0.047 | 0.710 ± 0.045 |
| | 12 - 18 mo | 0.669 ± 0.037 | **0.730 ± 0.031** | 0.678 ± 0.056 | 0.663 ± 0.052 | 0.643 ± 0.070 |
| | 18 - 24 mo | 0.632 ± 0.037 | **0.658 ± 0.056** | 0.594 ± 0.047 | 0.602 ± 0.068 | 0.552 ± 0.077 |
| $EDSS_{mean}$ As Severity Category | 0 - 6 mo | **0.782 ± 0.028** | 0.688 ± 0.009 | 0.566 ± 0.028 | 0.580 ± 0.029 | 0.569 ± 0.036 |
| | 6 - 12 mo | **0.709 ± 0.044** | 0.672 ± 0.016 | 0.601 ± 0.024 | 0.596 ± 0.021 | 0.555 ± 0.038 |
| | 12 - 18 mo | **0.674 ± 0.037** | 0.648 ± 0.022 | 0.574 ± 0.027 | 0.589 ± 0.028 | 0.538 ± 0.032 |
| | 18 - 24 mo | **0.632 ± 0.037** | 0.616 ± 0.025 | 0.540 ± 0.018 | 0.557 ± 0.031 | 0.513 ± 0.055 |

Table 2: Ablation of the effects of standardization and interpolation of features for Neural CDE on MSOAC. All metrics are AUPRC, except for the $EDSS_{mean}$ task, where we use RMSE.

| Prediction Task | Prediction Window | W Standardize W Interpolate | W Standardize WO Interpolate | WO Standardize W Interpolate | WO Standardize WO Interpolate |
|---|---|---|---|---|---|
| $EDSS_{mean}$ | 0 - 6 mo | 1.408 ± 0.059 | **1.406 ± 0.072** | 4.508 ± 2.124 | 3.463 ± 1.142 |
| | 6 - 12 mo | **1.627 ± 0.055** | 1.633 ± 0.075 | 3.561 ± 1.289 | 3.677 ± 1.718 |
| | 12 - 18 mo | 1.652 ± 0.066 | **1.638 ± 0.071** | 3.633 ± 1.099 | 3.797 ± 1.246 |
| | 18 - 24 mo | 1.587 ± 0.078 | **1.578 ± 0.073** | 3.976 ± 2.341 | 4.047 ± 1.886 |
| $EDSS_{mean}$ > 3 | 0 - 6 mo | **0.908 ± 0.010** | **0.908 ± 0.013** | 0.729 ± 0.071 | 0.732 ± 0.078 |
| | 6 - 12 mo | 0.852 ± 0.016 | **0.853 ± 0.022** | 0.662 ± 0.060 | 0.697 ± 0.033 |
| | 12 - 18 mo | 0.797 ± 0.027 | **0.798 ± 0.025** | 0.595 ± 0.087 | 0.619 ± 0.107 |
| | 18 - 24 mo | **0.742 ± 0.035** | **0.742 ± 0.037** | 0.571 ± 0.117 | 0.564 ± 0.071 |
| $EDSS_{mean}$ > 5 | 0 - 6 mo | **0.872 ± 0.021** | 0.868 ± 0.022 | 0.587 ± 0.097 | 0.612 ± 0.090 |
| | 6 - 12 mo | 0.793 ± 0.039 | **0.799 ± 0.036** | 0.560 ± 0.112 | 0.481 ± 0.114 |
| | 12 - 18 mo | 0.730 ± 0.031 | **0.731 ± 0.040** | 0.441 ± 0.117 | 0.438 ± 0.147 |
| | 18 - 24 mo | **0.658 ± 0.056** | **0.658 ± 0.052** | 0.380 ± 0.126 | 0.397 ± 0.141 |
| $EDSS_{mean}$ As Severity Category | 0 - 6 mo | **0.688 ± 0.009** | **0.688 ± 0.018** | 0.437 ± 0.082 | 0.413 ± 0.082 |
| | 6 - 12 mo | **0.672 ± 0.016** | 0.670 ± 0.019 | 0.448 ± 0.069 | 0.438 ± 0.080 |
| | 12 - 18 mo | **0.648 ± 0.022** | **0.648 ± 0.022** | 0.438 ± 0.050 | 0.441 ± 0.074 |
| | 18 - 24 mo | **0.616 ± 0.025** | **0.616 ± 0.020** | 0.464 ± 0.049 | 0.453 ± 0.058 |

**Interpolation Scheme Ablation.** Here we investigate how the performance changes when using different interpolation schemes. For this we standardize the features and take the best performing models for each scheme. This also tests the two methods for enforcing physical solutions - the method of reprocessing the dataset is contained in the Linear, Hermite Cubic, Mono Cubic and Natural Cubic interpolations, the recti

Table 3: Ablation of which interpolations work best for Neural CDE on MSOAC. All metrics are AUPRC, except for the $EDSS_{mean}$ task, where we use RMSE.

| Prediction Task | Prediction Window | Linear | Natural Cubic | Hermite Cubic | Monotonic Cubic | Recti-Linear | Recti-Cubic |
|---|---|---|---|---|---|---|---|
| $EDSS_{mean}$ | 0 - 6 mo | $1.409 \pm 0.077$ | $1.409 \pm 0.067$ | $1.409 \pm 0.059$ | $1.411 \pm 0.072$ | $1.418 \pm 0.062$ | $\mathbf{1.408 \pm 0.059}$ |
| | 6 - 12 mo | $\mathbf{1.627 \pm 0.055}$ | $1.633 \pm 0.091$ | $1.644 \pm 0.080$ | $1.644 \pm 0.077$ | $1.639 \pm 0.062$ | $1.651 \pm 0.100$ |
| | 12 - 18 mo | $1.672 \pm 0.079$ | $1.671 \pm 0.063$ | $1.673 \pm 0.079$ | $\mathbf{1.652 \pm 0.066}$ | $1.663 \pm 0.087$ | $1.676 \pm 0.054$ |
| | 18 - 24 mo | $1.602 \pm 0.089$ | $1.597 \pm 0.067$ | $\mathbf{1.587 \pm 0.078}$ | $1.592 \pm 0.060$ | $1.608 \pm 0.090$ | $1.589 \pm 0.080$ |
| $EDSS_{mean}$ > 3 | 0 - 6 mo | $0.902 \pm 0.012$ | $0.903 \pm 0.014$ | $0.903 \pm 0.011$ | $0.903 \pm 0.010$ | $\mathbf{0.908 \pm 0.010}$ | $0.904 \pm 0.012$ |
| | 6 - 12 mo | $0.847 \pm 0.016$ | $0.848 \pm 0.018$ | $0.850 \pm 0.017$ | $0.848 \pm 0.020$ | $0.846 \pm 0.021$ | $\mathbf{0.852 \pm 0.016}$ |
| | 12 - 18 mo | $0.794 \pm 0.027$ | $0.795 \pm 0.027$ | $\mathbf{0.797 \pm 0.033}$ | $\mathbf{0.797 \pm 0.027}$ | $0.792 \pm 0.024$ | $0.795 \pm 0.024$ |
| | 18 - 24 mo | $0.733 \pm 0.042$ | $0.740 \pm 0.042$ | $0.737 \pm 0.036$ | $0.738 \pm 0.036$ | $\mathbf{0.742 \pm 0.035}$ | $0.738 \pm 0.042$ |
| $EDSS_{mean}$ > 5 | 0 - 6 mo | $0.866 \pm 0.018$ | $0.865 \pm 0.032$ | $0.869 \pm 0.018$ | $0.864 \pm 0.022$ | $\mathbf{0.872 \pm 0.021}$ | $0.865 \pm 0.018$ |
| | 6 - 12 mo | $0.788 \pm 0.030$ | $0.789 \pm 0.033$ | $0.790 \pm 0.026$ | $0.788 \pm 0.025$ | $0.791 \pm 0.032$ | $\mathbf{0.793 \pm 0.039}$ |
| | 12 - 18 mo | $0.724 \pm 0.034$ | $\mathbf{0.730 \pm 0.031}$ | $0.728 \pm 0.027$ | $0.727 \pm 0.044$ | $0.726 \pm 0.039$ | $0.722 \pm 0.030$ |
| | 18 - 24 mo | $\mathbf{0.658 \pm 0.043}$ | $0.654 \pm 0.055$ | $0.647 \pm 0.052$ | $0.652 \pm 0.050$ | $\mathbf{0.658 \pm 0.056}$ | $\mathbf{0.658 \pm 0.042}$ |
| $EDSS_{mean}$ As Severity Category | 0 - 6 mo | $0.686 \pm 0.014$ | $0.683 \pm 0.016$ | $0.686 \pm 0.012$ | $0.686 \pm 0.013$ | $\mathbf{0.688 \pm 0.009}$ | $0.686 \pm 0.017$ |
| | 6 - 12 mo | $0.663 \pm 0.018$ | $0.669 \pm 0.017$ | $\mathbf{0.672 \pm 0.016}$ | $0.664 \pm 0.015$ | $0.668 \pm 0.016$ | $0.666 \pm 0.025$ |
| | 12 - 18 mo | $0.638 \pm 0.020$ | $\mathbf{0.648 \pm 0.022}$ | $0.644 \pm 0.024$ | $0.643 \pm 0.026$ | $0.645 \pm 0.020$ | $0.643 \pm 0.018$ |
| | 18 - 24 mo | $\mathbf{0.616 \pm 0.021}$ | $\mathbf{0.616 \pm 0.025}$ | $0.612 \pm 0.023$ | $0.613 \pm 0.018$ | $0.614 \pm 0.028$ | $0.614 \pm 0.027$ |

method is contained in the Rectilinear and Recticubic interpolations. The values for the Neural CDE are given in Table 3. We see that there is essentially no difference between the interpolation choices. Each one performs best on a task, and all values are within a standard deviation of each other. We see the same results for ODE-RNN, Latent ODE and GRU-ODE whose results are given in Appendix C. This is consistent with the findings in (Morrill et al., 2021) where the different interpolation schemes perform as well as each other since the learnable part of the Neural CDE makes up for the different interpolations.

## 6 Conclusion

We investigated the use of continuous time models to predict the progression of multiple sclerosis from performance outcome measures and demographic data. We compared four continuous time models against the previous best discrete time model - temporal convolutional network (TCN) - on a public multiple sclerosis dataset (MSOAC). We saw that Neural Controlled Differential Equations are able to beat TCN on the majority of experiments (ten out of sixteen), showing a well tuned continuous time model is able to outperform or compete with the current best discrete time model. This thorough investigation shows adjusting the modeling choices leads to changes in experimental results and lays the foundation for further research into the application of continuous time models for MS progression. After extensive ablation studies we found that standardization is vital to achieve competitive results using the Neural CDE, whereas the exact interpolation scheme is not. To adapt the framework of Roy et al. (2022) we also introduced two new control signals and a way to efficiently batch irregular time-series via these control signals as secondary contributions.

**Limitations.** We saw that three of the four continuous time models (ODE-RNN, Latent ODE, GRU-ODE) were not able to consistently outperform TCN, demonstrating modeling in continuous time does not guarantee performance gains, and that choosing the correct model is still a crucial step in the prediction pipeline. We also saw that continuous time models required a feature selection step in order to reach their peak performance, adding further resource demands to the training.

**Future Work.** The main aim of this work was to demonstrate that continuous time models can be a competitive option for modeling MS progression. We achieved this on Neural CDEs, and so our next steps

are to continue to test Neural CDEs on more MS datasets such as Floodlight (Baker et al., 2021) or MS Mosaic (Roy et al., 2023). We also plan to investigate how the Neural CDE scales to more features, and making it more stable in this scenario. From an engineering standpoint, we plan to improve the interpolation speed by parallelizing and optimizing the interpolation process. Finally, in light of the objective of clinicians to limit progression of MS, we plan to include intervention modeling in our setup.

**Broader Impact Statement**

In our work we investigate using continuous time models to model the progression of Multiple Sclerosis. Our work has shown that in some scenarios Neural Controlled Differential Equations can outperform the current best performing discrete time model - Temporal Convolutional Network - for longitudinal prediction.

**Applications.** We view positive applications of our work, since this could be used to help practitioners treat Multiple Sclerosis and also demonstrates for future research that continuous time models can be effective for this problem. Since this work is in early stages it *must* not be deployed yet, and more research and evaluation is required first: a tool of this nature needs to be tested to the same level of scrutiny as drug trials. If deployed, it *must* be used alongside the expert opinion of trained practitioners, and if in disagreement a further expert assessment is sought out. Crucially, in its current state a trained medical practitioner's judgement is more reliable than the Neural CDE model.

**Datasets.** We use a publicly available dataset in this work: Multiple Sclerosis Outcome Assessments Consortium (MSOAC) (Rudick et al., 2014) (`https://c-path.org/programs/msoac/`). Since the dataset is publicly available Institutional Review Board approval is not required.

**Code Release**

Our code does not exist in isolation but as part of a larger code base containing proprietary code. As such our code is not publicly available at this time, we plan to open-source it in the future. We have included implementation details in the Appendix, including library specific caveats in Appendix G and their fixes to aid reproduction of our results.

**Author Contributions**

**Alexander Norcliffe:** Lead author. Implemented the continuous models in TensorFlow. Implemented the interpolation schemes. Conceptualised and implemented the continuous model specific preprocessing. Implemented the irregular batching and wrote the proofs in the Appendix. Lead the writing of the paper and author responses.

**Lev Proleev:** Ran the experiments. Provided significant conceptual input. Provided significant assistance implementing continuous models and continuous preprocessing. Implemented the general preprocessing pipeline as part of (Roy et al., 2022). Reviewed code. Contributed to the writing of the paper.

**Diana Mincu:** Provided assistance implementing continuous models and continuous preprocessing. Provided significant conceptual input. Implemented the general preprocessing pipeline as part of (Roy et al., 2022). Reviewed code. Contributed to the writing of the paper and the author responses.

**Fletcher Lee Hartsell:** As an expert in Multiple Sclerosis, provided very specialised input during project meetings.

**Katherine Heller:** Provided input during project meetings.

**Subhrajit Roy:** Senior Author. Conceptualisation of the use of continuous methods in the Multiple Sclerosis setting. Lead project to completion. Provided significant conceptual input. Provided assistance implementing continuous models and continuous preprocessing. Implemented the general preprocessing pipeline as part of (Roy et al., 2022). Reviewed code. Contributed to the writing of the paper and the author responses.

**Acknowledgments**

We would like to thank the anonymous reviewers and action editor Qibin Zhao for their time and effort to review the paper. We'd like to thank Patrick Kidger for introducing Alexander Norcliffe to Subhrajit Roy

which helped to start this project. We'd like to thank James Morrill and Patrick Kidger for offering guidance on the use of different control signals in Neural CDEs. We'd like to thank our colleague Mercy Asiedu for providing a detailed review of the paper before submission. We'd also like to thank our colleague Wenhui Hu for help reviewing code. We'd finally like to thank our colleagues Emily Salkey, Eltayeb Ahmed and Chintan Ghate for interesting discussion.

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

## A    Additional RNN Baseline

Here we present results for a recurrent neural network (RNN) (Hopfield, 1984) on the MSOAC data, as an additional deep learning baseline deisgned for sequences not tested in Roy et al. (2022). The RNN consists of a ReLU multilayer perceptron that acts on a hidden state and observation, so that the hidden state at observation $n+1$ is given by $\boldsymbol{h}_{n+1} = f_\theta^c(\boldsymbol{h}_n, \boldsymbol{x}_{n+1})$, with $\boldsymbol{h}_{-1} = \boldsymbol{0}$. Predictions at time $t_n$ are made by taking the hidden state, using a learnable prediction linear layer on it, and applying a final activation relevant to the task, $\hat{y}(t_n) = d_\phi(\boldsymbol{h}_n)$. We train until convergence using the Adam optimizer with a batchsize of 1 (as is done for TCN in Roy et al. (2022)) and learning rate 0.001. We vary the number of hidden layers in the MLP, and the width of the MLP hidden layers, which is also the width of the hidden state. Hyperparameters are found using a grid search with 10 fold cross-validation (listed in Appendix B), the 10 models trained on these folds of training data are evaluated to obtain test metric mean and standard deviations. We give the results for RNN, Neural CDE and TCN in Table 4.

Table 4: RNN and NCDE MSOAC values against the best performing model from (Roy et al., 2022) (TCN). All metrics are AUPRC, except for the EDSS$_{\text{mean}}$ task, where we use RMSE. Best values are in **bold** and values better than TCN but not best are underlined.

| Prediction Task | Prediction Window | TCN | Neural CDE | RNN |
|---|---|---|---|---|
| EDSS$_{\text{mean}}$ | 0 -  6 mo | **1.264 ± 0.055** | 1.408 ± 0.059 | 1.318 ± 0.049 |
| | 6 - 12 mo | 1.650 ± 0.067 | **1.627 ± 0.055** | 1.681 ± 0.068 |
| | 12 - 18 mo | 1.725 ± 0.074 | **1.652 ± 0.066** | 1.801 ± 0.044 |
| | 18 - 24 mo | 1.666 ± 0.128 | **1.587 ± 0.078** | 1.691 ± 0.083 |
| EDSS$_{\text{mean}}$ > 3 | 0 -  6 mo | **0.909 ± 0.014** | 0.908 ± 0.010 | 0.898 ± 0.010 |
| | 6 - 12 mo | 0.820 ± 0.027 | **0.852 ± 0.016** | 0.818 ± 0.023 |
| | 12 - 18 mo | 0.768 ± 0.031 | **0.797 ± 0.027** | 0.744 ± 0.036 |
| | 18 - 24 mo | 0.703 ± 0.038 | **0.742 ± 0.035** | 0.704 ± 0.030 |
| EDSS$_{\text{mean}}$ > 5 | 0 -  6 mo | 0.848 ± 0.035 | **0.872 ± 0.021** | 0.856 ± 0.022 |
| | 6 - 12 mo | 0.722 ± 0.039 | **0.793 ± 0.039** | 0.738 ± 0.031 |
| | 12 - 18 mo | 0.669 ± 0.037 | **0.730 ± 0.031** | 0.560 ± 0.023 |
| | 18 - 24 mo | 0.632 ± 0.037 | **0.658 ± 0.056** | 0.472 ± 0.076 |
| EDSS$_{\text{mean}}$ As Severity Category | 0 -  6 mo | **0.782 ± 0.028** | 0.688 ± 0.009 | 0.704 ± 0.013 |
| | 6 - 12 mo | **0.709 ± 0.044** | 0.672 ± 0.016 | 0.638 ± 0.017 |
| | 12 - 18 mo | **0.674 ± 0.037** | 0.648 ± 0.022 | 0.600 ± 0.026 |
| | 18 - 24 mo | **0.632 ± 0.037** | 0.616 ± 0.025 | 0.577 ± 0.018 |

We see that RNN is rarely able to beat TCN, but it does happen. However, RNN is never the best performing model. To quantify this, if we take the rating across the 16 tasks (1 being best performing, 3 being worst), the average ratings for TCN, Neural CDE and RNN are $1.83 \pm 0.73$, $1.50 \pm 0.71$ and $2.69 \pm 0.46$ respectively. This shows that Neural CDE is the best performing, followed by TCN, and then RNN. Crucially it further supports the use of Neural CDEs in this context.

## B    Implementation and Training Details

All models are implemented using TensorFlow (Abadi et al., 2015) and Keras (Chollet et al., 2015) using the TensorFlow ODE solver `https://www.tensorflow.org/probability/api_docs/python/tfp/math/ode`. Our code is not publicly available currently, however, we plan to open-source in the future and include full implementation details here for reproducibility purposes. All models are trained for 50 epochs with the Adam optimizer (Kingma & Ba, 2015). Hyperparameters are found using a grid search with 10 fold cross-validation. We found two library specific caveats with the TensorFlow ODE solver, we describe them and provide the fixes in Appendix G.

**B.1 Hyperparameters**

Here we give the different hyperparameters of each model, the values used in the grid search and the final value found. These are given for Neural CDE in Table 5, ODE-RNN in Table 6, Latent ODE in Table 7, GRU-ODE in Table 8 and RNN in Table 9. In some cases the selected hyperparameter varied between the tasks, in particular for latent widths and MLP hidden widths. We record these specific values in Table 10.

Table 5: Table of hyperparameters for Neural CDE.

| Hyperparameter | Candidate Values | Best Value |
|---|---|---|
| Embedding Layer Learning Rate | 0.001 | 0.001 |
| Embedding Layer Hidden Width | [10, 20, 40] | Varies |
| Embedding Layer No. Hidden Layers | 1 | 1 |
| NCDE Learning Rate | 0.001 | 0.001 |
| NCDE Hidden Width | [10, 20, 40] | Varies |
| NCDE No. Hidden Layers | 2 | 2 |
| Prediction Layer Learning Rate | 0.01 | 0.01 |
| Latent Size | [5, 10, 20] | Varies |
| Batchsize | [50, 100, 200] | 100 |

Table 6: Table of hyperparameters for ODE-RNN.

| Hyperparameter | Candidate Values | Best Value |
|---|---|---|
| Discontinuous Update Learning Rate | 0.001 | 0.001 |
| Discontinuous Update Hidden Width | [10, 20, 40] | Varies |
| Discontinuous Update No. Hidden Layers | 1 | 1 |
| ODE Learning Rate | 0.001 | 0.001 |
| ODE Hidden Width | [10, 20, 40] | Varies |
| ODE No. Hidden Layers | 2 | 2 |
| Prediction Layer Learning Rate | 0.01 | 0.01 |
| Latent Size | [5, 10, 20] | Varies |
| Batchsize | [50, 100, 200] | 100 |

Table 7: Table of hyperparameters for Latent ODE.

| Hyperparameter | Candidate Values | Best Value |
|---|---|---|
| ODE-RNN Encoder Discontinuous Update Learning Rate | 0.001 | 0.001 |
| ODE-RNN Encoder Discontinuous Update Hidden Width | [10, 20, 40] | Varies |
| ODE-RNN Encoder Discontinuous Update No. Hidden Layers | 1 | 1 |
| ODE-RNN Encoder ODE Learning Rate | 0.001 | 0.001 |
| ODE-RNN Encoder ODE Hidden Width | [10, 20, 40] | Varies |
| ODE-RNN Encoder ODE No. Hidden Layers | 2 | 2 |
| ODE Decoder Learning Rate | 0.001 | 0.001 |
| ODE Decoder Hidden Width | [10, 20, 40] | Varies |
| ODE Decoder No. Hidden Layers | 2 | 2 |
| Prediction Layer Learning Rate | 0.01 | 0.01 |
| Latent Size | [5, 10, 20] | Varies |
| Batchsize | [50, 100, 200] | 100 |

Table 8: Table of hyperparameters for GRU-ODE.

| Hyperparameter | Candidate Values | Best Value |
|---|---|---|
| Embedding Layer Learning Rate | 0.001 | 0.001 |
| Embedding Layer Hidden Width | [10, 20, 40] | Varies |
| Embedding Layer No. Hidden Layers | 2 | 2 |
| GRU-ODE Learning Rate | 0.001 | 0.001 |
| Prediction Layer Learning Rate | 0.01 | 0.01 |
| Latent Size | [5, 10, 20] | Varies |
| Batchsize | [50, 100, 200] | 100 |

Table 9: Table of hyperparameters for the RNN.

| Hyperparameter | Candidate Values | Best Value |
|---|---|---|
| Hidden Width | [32, 64] | Varies |
| No. Hidden Layers | [2, 3] | Varies |

Table 10: MSOAC Hyperparameters that varied between tasks. For the continuous models, the first value is the selected Latent Width and the second value is the selected Hidden Width of the models. For the RNN, the first value is the hidden width, the second value is the number of hidden layers.

| Prediction Task | Prediction Window | Neural CDE | | ODE RNN | | Latent ODE | | GRU ODE | | RNN | |
|---|---|---|---|---|---|---|---|---|---|---|---|
| EDSS$_{mean}$ | 0 - 6 mo | 40, | 20 | 40, | 10 | 40, | 5 | 40, | 5 | 64, | 2 |
| | 6 - 12 mo | 40, | 20 | 40, | 5 | 40, | 5 | 40, | 20 | 64, | 2 |
| | 12 - 18 mo | 40, | 20 | 40, | 5 | 40, | 5 | 40, | 20 | 64, | 2 |
| | 18 - 24 mo | 40, | 10 | 40, | 5 | 40, | 10 | 40, | 10 | 64, | 2 |
| EDSS$_{mean}$ > 3 | 0 - 6 mo | 40, | 20 | 20, | 10 | 10, | 20 | 40, | 20 | 32, | 2 |
| | 6 - 12 mo | 20, | 20 | 20, | 10 | 40, | 5 | 40, | 20 | 64, | 2 |
| | 12 - 18 mo | 40, | 20 | 20, | 5 | 10, | 20 | 40, | 20 | 64, | 2 |
| | 18 - 24 mo | 20, | 20 | 40, | 5 | 10, | 20 | 40, | 20 | 32, | 2 |
| EDSS$_{mean}$ > 5 | 0 - 6 mo | 40, | 20 | 10, | 10 | 10, | 10 | 40, | 20 | 32, | 3 |
| | 6 - 12 mo | 40, | 20 | 40, | 5 | 20, | 20 | 40, | 20 | 64, | 2 |
| | 12 - 18 mo | 40, | 20 | 40, | 5 | 10, | 20 | 40, | 20 | 64, | 2 |
| | 18 - 24 mo | 40, | 20 | 10, | 20 | 10, | 20 | 40, | 20 | 64, | 2 |
| EDSS$_{mean}$ As Severity Category | 0 - 6 mo | 40, | 20 | 40, | 5 | 40, | 10 | 40, | 20 | 64, | 2 |
| | 6 - 12 mo | 40, | 20 | 40, | 5 | 40, | 5 | 40, | 20 | 64, | 2 |
| | 12 - 18 mo | 40, | 20 | 40, | 5 | 10, | 20 | 40, | 20 | 64, | 2 |
| | 18 - 24 mo | 40, | 20 | 20, | 10 | 10, | 20 | 40, | 20 | 64, | 2 |

## C Ablation Studies

In Section 5 we carried out ablations investigating how standardization, interpolating missing values and interpolation scheme affect the performance of Neural CDEs. In this section we do the same for ODE-RNN, Latent ODE and GRU-ODE.

### C.1 Standardizing and Interpolating

We look at the effects of standardizing features and interpolating missing features for ODE-RNN in Table 11, Latent ODE in Table 12 and GRU-ODE in Table 13. We see that on the whole ODE-RNN does best when standardizing features and filling missing values with zeros, Latent ODE does not have a clear superior scheme and GRU-ODE does best when standardizing features and filling missing values with interpolated values. However for these three methods the differences are small, whereas with Neural CDEs there was a clear distinction between standardizing and not standardizing features. This is likely due to the models' dynamics functions. The dynamics function for the Neural CDE ends with $dX/ds$, whereas all others end with $d\psi/ds$. In MSOAC the true times are fairly small and therefore so is $d\psi/ds$. However, $X$ can be quite large, and therefore so is $dX/ds$. And so standardizing $X$ aids Neural CDEs more because $dX/ds$ in the dynamics is explicitly shifted to a more manageable range, whereas $d\psi/ds$ was already in that range.

Table 11: Ablation of the effects of standardization and interpolation of features for ODE-RNN on MSOAC. All metrics are AUPRC, except for the $\text{EDSS}_{\text{mean}}$ task, where we use RMSE.

| Prediction Task | Prediction Window | W Standardize W Interpolate | W Standardize WO Interpolate | WO Standardize W Interpolate | WO Standardize WO Interpolate |
|---|---|---|---|---|---|
| $\text{EDSS}_{\text{mean}}$ | 0 - 6 mo | $1.768 \pm 0.139$ | $1.743 \pm 0.145$ | $\mathbf{1.725 \pm 0.167}$ | $1.756 \pm 0.138$ |
| | 6 - 12 mo | $1.886 \pm 0.248$ | $1.870 \pm 0.103$ | $1.842 \pm 0.167$ | $\mathbf{1.821 \pm 0.163}$ |
| | 12 - 18 mo | $1.826 \pm 0.123$ | $\mathbf{1.802 \pm 0.075}$ | $1.825 \pm 0.117$ | $1.827 \pm 0.086$ |
| | 18 - 24 mo | $1.741 \pm 0.100$ | $\mathbf{1.709 \pm 0.101}$ | $1.717 \pm 0.066$ | $1.727 \pm 0.100$ |
| $\text{EDSS}_{\text{mean}}$ $> 3$ | 0 - 6 mo | $0.869 \pm 0.018$ | $\mathbf{0.873 \pm 0.009}$ | $0.866 \pm 0.026$ | $0.864 \pm 0.025$ |
| | 6 - 12 mo | $0.823 \pm 0.024$ | $\mathbf{0.825 \pm 0.029}$ | $0.815 \pm 0.030$ | $0.816 \pm 0.027$ |
| | 12 - 18 mo | $0.766 \pm 0.032$ | $0.762 \pm 0.034$ | $0.758 \pm 0.051$ | $\mathbf{0.768 \pm 0.023}$ |
| | 18 - 24 mo | $0.697 \pm 0.044$ | $0.695 \pm 0.042$ | $\mathbf{0.701 \pm 0.038}$ | $0.698 \pm 0.032$ |
| $\text{EDSS}_{\text{mean}}$ $> 5$ | 0 - 6 mo | $0.807 \pm 0.032$ | $\mathbf{0.818 \pm 0.027}$ | $0.792 \pm 0.054$ | $0.806 \pm 0.027$ |
| | 6 - 12 mo | $0.751 \pm 0.026$ | $\mathbf{0.757 \pm 0.040}$ | $0.726 \pm 0.028$ | $0.728 \pm 0.049$ |
| | 12 - 18 mo | $0.678 \pm 0.056$ | $\mathbf{0.682 \pm 0.035}$ | $0.655 \pm 0.029$ | $0.662 \pm 0.050$ |
| | 18 - 24 mo | $0.594 \pm 0.047$ | $\mathbf{0.600 \pm 0.044}$ | $0.588 \pm 0.053$ | $0.593 \pm 0.055$ |
| $\text{EDSS}_{\text{mean}}$ As Severity Category | 0 - 6 mo | $0.566 \pm 0.028$ | $\mathbf{0.578 \pm 0.024}$ | $0.564 \pm 0.025$ | $0.564 \pm 0.038$ |
| | 6 - 12 mo | $\mathbf{0.601 \pm 0.024}$ | $0.593 \pm 0.036$ | $0.588 \pm 0.018$ | $0.583 \pm 0.033$ |
| | 12 - 18 mo | $0.574 \pm 0.027$ | $0.576 \pm 0.026$ | $\mathbf{0.577 \pm 0.027}$ | $\mathbf{0.577 \pm 0.026}$ |
| | 18 - 24 mo | $0.540 \pm 0.018$ | $\mathbf{0.545 \pm 0.034}$ | $0.542 \pm 0.028$ | $0.542 \pm 0.022$ |

Table 12: Ablation of the effects of standardization and interpolation of features for Latent ODE on MSOAC. All metrics are AUPRC, except for the $EDSS_{mean}$ task, where we use RMSE.

| Prediction Task | Prediction Window | W Standardize W Interpolate | W Standardize WO Interpolate | WO Standardize W Interpolate | WO Standardize WO Interpolate |
|---|---|---|---|---|---|
| $EDSS_{mean}$ | 0 - 6 mo | $1.617 \pm 0.072$ | $1.615 \pm 0.088$ | $\mathbf{1.585 \pm 0.079}$ | $1.596 \pm 0.094$ |
| | 6 - 12 mo | $1.776 \pm 0.071$ | $1.784 \pm 0.081$ | $\mathbf{1.754 \pm 0.099}$ | $1.775 \pm 0.073$ |
| | 12 - 18 mo | $1.810 \pm 0.090$ | $1.763 \pm 0.094$ | $\mathbf{1.735 \pm 0.100}$ | $1.758 \pm 0.084$ |
| | 18 - 24 mo | $1.707 \pm 0.097$ | $1.683 \pm 0.094$ | $\mathbf{1.644 \pm 0.090}$ | $1.653 \pm 0.074$ |
| $EDSS_{mean}$ > 3 | 0 - 6 mo | $0.871 \pm 0.017$ | $\mathbf{0.879 \pm 0.017}$ | $0.866 \pm 0.025$ | $0.867 \pm 0.018$ |
| | 6 - 12 mo | $0.819 \pm 0.030$ | $\mathbf{0.824 \pm 0.026}$ | $0.818 \pm 0.034$ | $0.819 \pm 0.020$ |
| | 12 - 18 mo | $0.776 \pm 0.027$ | $0.770 \pm 0.040$ | $\mathbf{0.777 \pm 0.037}$ | $\mathbf{0.777 \pm 0.024}$ |
| | 18 - 24 mo | $0.703 \pm 0.044$ | $\mathbf{0.713 \pm 0.044}$ | $0.711 \pm 0.032$ | $0.712 \pm 0.035$ |
| $EDSS_{mean}$ > 5 | 0 - 6 mo | $\mathbf{0.788 \pm 0.026}$ | $0.786 \pm 0.031$ | $0.786 \pm 0.025$ | $0.775 \pm 0.037$ |
| | 6 - 12 mo | $0.740 \pm 0.047$ | $0.737 \pm 0.035$ | $0.732 \pm 0.039$ | $\mathbf{0.741 \pm 0.037}$ |
| | 12 - 18 mo | $0.663 \pm 0.052$ | $0.665 \pm 0.049$ | $0.676 \pm 0.040$ | $\mathbf{0.679 \pm 0.035}$ |
| | 18 - 24 mo | $0.602 \pm 0.068$ | $\mathbf{0.613 \pm 0.064}$ | $0.602 \pm 0.058$ | $0.604 \pm 0.063$ |
| $EDSS_{mean}$ As Severity Category | 0 - 6 mo | $0.580 \pm 0.029$ | $0.590 \pm 0.020$ | $0.585 \pm 0.018$ | $\mathbf{0.591 \pm 0.034}$ |
| | 6 - 12 mo | $0.596 \pm 0.021$ | $0.599 \pm 0.018$ | $0.600 \pm 0.032$ | $\mathbf{0.609 \pm 0.018}$ |
| | 12 - 18 mo | $0.589 \pm 0.028$ | $0.593 \pm 0.026$ | $0.599 \pm 0.026$ | $\mathbf{0.610 \pm 0.024}$ |
| | 18 - 24 mo | $0.557 \pm 0.031$ | $0.563 \pm 0.034$ | $\mathbf{0.573 \pm 0.025}$ | $0.571 \pm 0.024$ |

Table 13: Ablation of the effects of standardization and interpolation of features for GRU-ODE on MSOAC. All metrics are AUPRC, except for the $EDSS_{mean}$ task, where we use RMSE.

| Prediction Task | Prediction Window | W Standardize W Interpolate | W Standardize WO Interpolate | WO Standardize W Interpolate | WO Standardize WO Interpolate |
|---|---|---|---|---|---|
| $EDSS_{mean}$ | 0 - 6 mo | $2.312 \pm 0.271$ | $2.295 \pm 0.362$ | $2.115 \pm 0.374$ | $\mathbf{2.043 \pm 0.188}$ |
| | 6 - 12 mo | $2.236 \pm 0.207$ | $2.252 \pm 0.183$ | $2.044 \pm 0.135$ | $\mathbf{1.977 \pm 0.119}$ |
| | 12 - 18 mo | $2.066 \pm 0.086$ | $2.066 \pm 0.138$ | $2.009 \pm 0.112$ | $\mathbf{2.001 \pm 0.096}$ |
| | 18 - 24 mo | $\mathbf{1.834 \pm 0.107}$ | $1.842 \pm 0.116$ | $1.880 \pm 0.069$ | $1.908 \pm 0.097$ |
| $EDSS_{mean}$ > 3 | 0 - 6 mo | $0.865 \pm 0.021$ | $\mathbf{0.867 \pm 0.019}$ | $0.848 \pm 0.024$ | $0.851 \pm 0.029$ |
| | 6 - 12 mo | $0.788 \pm 0.025$ | $\mathbf{0.791 \pm 0.023}$ | $0.780 \pm 0.033$ | $0.753 \pm 0.026$ |
| | 12 - 18 mo | $\mathbf{0.730 \pm 0.041}$ | $0.727 \pm 0.047$ | $0.657 \pm 0.040$ | $0.659 \pm 0.062$ |
| | 18 - 24 mo | $\mathbf{0.666 \pm 0.062}$ | $0.650 \pm 0.039$ | $0.577 \pm 0.089$ | $0.566 \pm 0.074$ |
| $EDSS_{mean}$ > 5 | 0 - 6 mo | $0.794 \pm 0.036$ | $\mathbf{0.805 \pm 0.031}$ | $0.756 \pm 0.035$ | $0.761 \pm 0.031$ |
| | 6 - 12 mo | $\mathbf{0.710 \pm 0.045}$ | $0.698 \pm 0.042$ | $0.671 \pm 0.061$ | $0.640 \pm 0.037$ |
| | 12 - 18 mo | $\mathbf{0.643 \pm 0.070}$ | $0.620 \pm 0.060$ | $0.531 \pm 0.132$ | $0.559 \pm 0.081$ |
| | 18 - 24 mo | $\mathbf{0.552 \pm 0.077}$ | $0.524 \pm 0.093$ | $0.425 \pm 0.164$ | $0.393 \pm 0.091$ |
| $EDSS_{mean}$ As Severity Category | 0 - 6 mo | $\mathbf{0.569 \pm 0.036}$ | $0.558 \pm 0.032$ | $0.510 \pm 0.021$ | $0.513 \pm 0.036$ |
| | 6 - 12 mo | $\mathbf{0.555 \pm 0.038}$ | $0.545 \pm 0.029$ | $0.492 \pm 0.033$ | $0.500 \pm 0.039$ |
| | 12 - 18 mo | $0.538 \pm 0.032$ | $\mathbf{0.548 \pm 0.038}$ | $0.496 \pm 0.043$ | $0.496 \pm 0.028$ |
| | 18 - 24 mo | $\mathbf{0.513 \pm 0.055}$ | $0.511 \pm 0.038$ | $0.442 \pm 0.033$ | $0.440 \pm 0.040$ |

## C.2 Interpolation Schemes

We investigate which interpolation schemes are best for ODE-RNN in Table 14, Latent ODE in Table 15 and GRU-ODE in Table 16. As was the case for Neural CDEs, we see that there is minimal difference between the choices of interpolation scheme.

Table 14: Ablation of which interpolations work best for ODE-RNN on MSOAC. All metrics are AUPRC, except for the $\text{EDSS}_{\text{mean}}$ task, where we use RMSE.

| Prediction Task | Prediction Window | Linear | Natural Cubic | Hermite Cubic | Monotonic Cubic | Recti-Linear | Recti-Cubic |
|---|---|---|---|---|---|---|---|
| $\text{EDSS}_{\text{mean}}$ | 0 - 6 mo | $1.805 \pm 0.186$ | $1.820 \pm 0.232$ | $1.939 \pm 0.290$ | $1.975 \pm 0.230$ | $1.889 \pm 0.294$ | $\mathbf{1.768 \pm 0.139}$ |
| | 6 - 12 mo | $1.898 \pm 0.204$ | $1.897 \pm 0.099$ | $1.917 \pm 0.196$ | $2.012 \pm 0.215$ | $1.915 \pm 0.186$ | $\mathbf{1.886 \pm 0.248}$ |
| | 12 - 18 mo | $\mathbf{1.826 \pm 0.123}$ | $1.896 \pm 0.174$ | $1.875 \pm 0.147$ | $1.855 \pm 0.128$ | $1.852 \pm 0.136$ | $1.835 \pm 0.143$ |
| | 18 - 24 mo | $\mathbf{1.741 \pm 0.100}$ | $1.754 \pm 0.108$ | $1.766 \pm 0.103$ | $1.801 \pm 0.136$ | $1.833 \pm 0.175$ | $1.757 \pm 0.136$ |
| $\text{EDSS}_{\text{mean}}$ > 3 | 0 - 6 mo | $0.863 \pm 0.020$ | $0.868 \pm 0.024$ | $0.864 \pm 0.024$ | $0.865 \pm 0.016$ | $0.863 \pm 0.017$ | $\mathbf{0.869 \pm 0.018}$ |
| | 6 - 12 mo | $0.816 \pm 0.025$ | $0.810 \pm 0.034$ | $\mathbf{0.823 \pm 0.024}$ | $0.820 \pm 0.027$ | $0.818 \pm 0.029$ | $0.814 \pm 0.026$ |
| | 12 - 18 mo | $0.747 \pm 0.028$ | $0.760 \pm 0.042$ | $0.756 \pm 0.027$ | $0.763 \pm 0.030$ | $\mathbf{0.766 \pm 0.032}$ | $0.752 \pm 0.026$ |
| | 18 - 24 mo | $0.695 \pm 0.032$ | $0.692 \pm 0.036$ | $0.673 \pm 0.038$ | $0.694 \pm 0.039$ | $0.692 \pm 0.034$ | $\mathbf{0.697 \pm 0.044}$ |
| $\text{EDSS}_{\text{mean}}$ > 5 | 0 - 6 mo | $0.803 \pm 0.030$ | $0.804 \pm 0.039$ | $\mathbf{0.807 \pm 0.032}$ | $\mathbf{0.807 \pm 0.027}$ | $0.802 \pm 0.028$ | $0.805 \pm 0.029$ |
| | 6 - 12 mo | $0.742 \pm 0.047$ | $0.732 \pm 0.056$ | $\mathbf{0.751 \pm 0.026}$ | $0.736 \pm 0.031$ | $0.730 \pm 0.060$ | $0.744 \pm 0.043$ |
| | 12 - 18 mo | $\mathbf{0.678 \pm 0.056}$ | $0.653 \pm 0.041$ | $0.672 \pm 0.053$ | $0.662 \pm 0.050$ | $0.675 \pm 0.040$ | $0.663 \pm 0.042$ |
| | 18 - 24 mo | $0.586 \pm 0.046$ | $0.592 \pm 0.045$ | $\mathbf{0.594 \pm 0.047}$ | $0.590 \pm 0.050$ | $0.581 \pm 0.049$ | $0.572 \pm 0.062$ |
| $\text{EDSS}_{\text{mean}}$ As Severity Category | 0 - 6 mo | $\mathbf{0.566 \pm 0.028}$ | $0.562 \pm 0.036$ | $0.556 \pm 0.040$ | $0.559 \pm 0.030$ | $0.558 \pm 0.024$ | $0.557 \pm 0.028$ |
| | 6 - 12 mo | $0.591 \pm 0.041$ | $0.579 \pm 0.034$ | $0.580 \pm 0.024$ | $0.583 \pm 0.046$ | $\mathbf{0.601 \pm 0.024}$ | $0.586 \pm 0.021$ |
| | 12 - 18 mo | $0.559 \pm 0.019$ | $0.560 \pm 0.042$ | $0.559 \pm 0.025$ | $\mathbf{0.574 \pm 0.027}$ | $0.570 \pm 0.020$ | $0.573 \pm 0.025$ |
| | 18 - 24 mo | $0.539 \pm 0.026$ | $0.538 \pm 0.029$ | $0.530 \pm 0.029$ | $0.538 \pm 0.030$ | $0.534 \pm 0.032$ | $\mathbf{0.540 \pm 0.018}$ |

Table 15: Ablation of which interpolations work best for Latent ODE on MSOAC. All metrics are AUPRC, except for the $\text{EDSS}_{\text{mean}}$ task, where we use RMSE. Recall that since Latent ODE encodes a whole sequence and then decodes to predict, the rectilinear and recticubic control signals cannot be used.

| Prediction Task | Prediction Window | Linear | Natural Cubic | Hermite Cubic | Monotonic Cubic |
|---|---|---|---|---|---|
| $\text{EDSS}_{\text{mean}}$ | 0 - 6 mo | $1.619 \pm 0.078$ | $1.620 \pm 0.095$ | $1.643 \pm 0.080$ | $\mathbf{1.617 \pm 0.072}$ |
| | 6 - 12 mo | $1.806 \pm 0.077$ | $\mathbf{1.776 \pm 0.071}$ | $1.819 \pm 0.132$ | $1.785 \pm 0.058$ |
| | 12 - 18 mo | $1.816 \pm 0.116$ | $\mathbf{1.810 \pm 0.090}$ | $1.810 \pm 0.089$ | $1.838 \pm 0.151$ |
| | 18 - 24 mo | $1.726 \pm 0.081$ | $1.717 \pm 0.094$ | $1.747 \pm 0.115$ | $\mathbf{1.707 \pm 0.097}$ |
| $\text{EDSS}_{\text{mean}}$ > 3 | 0 - 6 mo | $0.863 \pm 0.024$ | $0.866 \pm 0.017$ | $0.868 \pm 0.023$ | $\mathbf{0.871 \pm 0.017}$ |
| | 6 - 12 mo | $0.818 \pm 0.029$ | $0.817 \pm 0.019$ | $\mathbf{0.819 \pm 0.026}$ | $\mathbf{0.819 \pm 0.030}$ |
| | 12 - 18 mo | $\mathbf{0.776 \pm 0.027}$ | $0.769 \pm 0.022$ | $0.765 \pm 0.038$ | $0.763 \pm 0.033$ |
| | 18 - 24 mo | $0.699 \pm 0.054$ | $0.700 \pm 0.047$ | $0.692 \pm 0.045$ | $\mathbf{0.703 \pm 0.044}$ |
| $\text{EDSS}_{\text{mean}}$ > 5 | 0 - 6 mo | $0.779 \pm 0.044$ | $\mathbf{0.788 \pm 0.026}$ | $0.780 \pm 0.057$ | $0.779 \pm 0.040$ |
| | 6 - 12 mo | $0.733 \pm 0.039$ | $0.719 \pm 0.037$ | $0.718 \pm 0.046$ | $\mathbf{0.740 \pm 0.047}$ |
| | 12 - 18 mo | $0.654 \pm 0.038$ | $0.651 \pm 0.048$ | $0.656 \pm 0.044$ | $\mathbf{0.663 \pm 0.052}$ |
| | 18 - 24 mo | $0.581 \pm 0.050$ | $0.574 \pm 0.061$ | $0.590 \pm 0.047$ | $\mathbf{0.602 \pm 0.068}$ |
| $\text{EDSS}_{\text{mean}}$ As Severity Category | 0 - 6 mo | $\mathbf{0.580 \pm 0.029}$ | $0.568 \pm 0.035$ | $0.573 \pm 0.030$ | $0.559 \pm 0.024$ |
| | 6 - 12 mo | $0.595 \pm 0.026$ | $0.587 \pm 0.023$ | $\mathbf{0.596 \pm 0.021}$ | $0.589 \pm 0.031$ |
| | 12 - 18 mo | $0.587 \pm 0.030$ | $\mathbf{0.589 \pm 0.028}$ | $0.586 \pm 0.033$ | $0.585 \pm 0.031$ |
| | 18 - 24 mo | $\mathbf{0.557 \pm 0.021}$ | $0.555 \pm 0.033$ | $\mathbf{0.557 \pm 0.031}$ | $0.556 \pm 0.033$ |

Table 16: Ablation of which interpolations work best for GRU-ODE on MSOAC. All metrics are AUPRC, except for the $EDSS_{mean}$ task, where we use RMSE.

| Prediction Task | Prediction Window | Linear | Natural Cubic | Hermite Cubic | Monotonic Cubic | Recti-Linear | Recti-Cubic |
|---|---|---|---|---|---|---|---|
| $EDSS_{mean}$ | 0 - 6 mo | **2.312 ± 0.271** | 2.421 ± 0.199 | 2.570 ± 0.393 | 2.381 ± 0.445 | 2.434 ± 0.430 | 2.512 ± 0.630 |
| | 6 - 12 mo | 2.335 ± 0.342 | 2.275 ± 0.167 | **2.236 ± 0.207** | 2.290 ± 0.247 | 2.324 ± 0.379 | 2.468 ± 0.313 |
| | 12 - 18 mo | 2.077 ± 0.183 | **2.066 ± 0.086** | 2.114 ± 0.130 | 2.148 ± 0.229 | 2.126 ± 0.161 | 2.241 ± 0.287 |
| | 18 - 24 mo | 1.863 ± 0.105 | **1.834 ± 0.107** | 1.870 ± 0.124 | 1.863 ± 0.094 | 1.884 ± 0.075 | 1.914 ± 0.127 |
| $EDSS_{mean}$ $> 3$ | 0 - 6 mo | 0.854 ± 0.031 | 0.860 ± 0.012 | 0.855 ± 0.038 | 0.855 ± 0.030 | **0.865 ± 0.021** | 0.847 ± 0.027 |
| | 6 - 12 mo | 0.785 ± 0.029 | 0.773 ± 0.038 | 0.776 ± 0.047 | 0.787 ± 0.027 | 0.774 ± 0.036 | **0.788 ± 0.025** |
| | 12 - 18 mo | 0.694 ± 0.062 | **0.730 ± 0.041** | 0.704 ± 0.096 | 0.723 ± 0.049 | 0.723 ± 0.042 | 0.663 ± 0.060 |
| | 18 - 24 mo | **0.666 ± 0.062** | 0.632 ± 0.112 | 0.631 ± 0.071 | 0.559 ± 0.163 | 0.661 ± 0.062 | 0.623 ± 0.104 |
| $EDSS_{mean}$ $> 5$ | 0 - 6 mo | 0.773 ± 0.052 | **0.794 ± 0.036** | 0.782 ± 0.029 | 0.761 ± 0.041 | 0.790 ± 0.043 | 0.778 ± 0.052 |
| | 6 - 12 mo | 0.673 ± 0.061 | 0.702 ± 0.054 | **0.710 ± 0.045** | 0.695 ± 0.034 | 0.649 ± 0.063 | 0.698 ± 0.038 |
| | 12 - 18 mo | 0.630 ± 0.063 | 0.624 ± 0.093 | 0.623 ± 0.082 | **0.643 ± 0.070** | 0.639 ± 0.046 | 0.571 ± 0.137 |
| | 18 - 24 mo | 0.511 ± 0.060 | 0.463 ± 0.184 | 0.523 ± 0.069 | **0.552 ± 0.077** | 0.476 ± 0.117 | 0.514 ± 0.117 |
| $EDSS_{mean}$ As Severity Category | 0 - 6 mo | 0.552 ± 0.037 | 0.552 ± 0.021 | 0.567 ± 0.036 | 0.560 ± 0.026 | **0.569 ± 0.036** | 0.559 ± 0.031 |
| | 6 - 12 mo | 0.541 ± 0.037 | 0.544 ± 0.028 | 0.531 ± 0.050 | 0.541 ± 0.033 | 0.540 ± 0.037 | **0.555 ± 0.038** |
| | 12 - 18 mo | **0.538 ± 0.032** | 0.498 ± 0.036 | 0.504 ± 0.062 | 0.534 ± 0.032 | 0.513 ± 0.059 | 0.509 ± 0.058 |
| | 18 - 24 mo | **0.513 ± 0.055** | 0.464 ± 0.057 | 0.484 ± 0.054 | 0.496 ± 0.069 | 0.509 ± 0.040 | 0.479 ± 0.049 |

## C.3 Feature Ablation

In Section 5, it was stated that we used a reduced feature set for the continuous models. We use Paced Auditory Serial Addition Test, Symbol Digit Modalities Test, Nine Hole Peg Test, Timed 25-Foot Walk, current EDSS score, Age and Sex. We focus on most of the functional tests, removing most of the demographics and all questionnaires. In order to evaluate these features against the full feature set, we test Neural CDE on the full feature set and the reduced set in Table 17, we see that Neural CDE is consistently better on the reduced feature set, and significantly so.

Table 17: Feature Ablation on Neural CDE. ∗: Not evaluated.

| Prediction Task | Prediction Window | Neural CDE Reduced Features | Neural CDE All Features |
|---|---|---|---|
| $EDSS_{mean}$ $> 3$ | 0 - 6 mo | **0.908 ± 0.010** | 0.756 ± 0.113 |
| | 6 - 12 mo | **0.852 ± 0.016** | 0.711 ± 0.089 |
| | 12 - 18 mo | **0.797 ± 0.027** | 0.695 ± 0.058 |
| | 18 - 24 mo | **0.742 ± 0.035** | 0.641 ± 0.068 |
| $EDSS_{mean}$ $> 5$ | 0 - 6 mo | **0.872 ± 0.021** | 0.609 ± 0.085 |
| | 6 - 12 mo | **0.793 ± 0.039** | 0.517 ± 0.112 |
| | 12 - 18 mo | **0.730 ± 0.031** | 0.455 ± 0.116 |
| | 18 - 24 mo | **0.658 ± 0.056** | 0.445 ± 0.122 |
| $EDSS_{mean}$ As Severity Category | 0 - 6 mo | **0.688 ± 0.009** | 0.487 ± 0.067 |
| | 6 - 12 mo | **0.672 ± 0.016** | 0.517 ± 0.065 |
| | 12 - 18 mo | **0.648 ± 0.022** | 0.482 ± 0.071 |
| | 18 - 24 mo | 0.616 ± 0.025 | ∗ |

We further hypothesize that one of the reasons NCDE becomes unstable to a large amount of features is because the dynamics function is given by a matrix-vector multiplication. The dynamics of the NCDE is given by $\frac{d\boldsymbol{h}}{ds} = f_\theta^c(\boldsymbol{h}, \psi(s))\frac{dX}{ds}$ which is a matrix-vector multiplication, whereas the other three models have a dynamics function given by $\frac{d\boldsymbol{h}}{ds} = f_\theta^c(\boldsymbol{h}, \psi(s))\frac{d\psi}{ds}$ which is a vector-scalar multiplication. This would mean, since all functions $f_\theta^c$ have a final tanh activation, the NCDE dynamics are bounded by $\pm\sum_i dX_i/ds$, whereas the other models dynamics are bounded by $\pm d\psi/ds$. This explains why NCDE is unstable to a large amount of features, because the magnitude of the velocity of the hidden state grows with the number of features. One possible solution would be to divide the NCDE dynamics by the dimensionality of the vector $X$, which would make the dynamics agnostic to the number of features. It further explains the importance of standardizing $X$, keeping $dX/ds$ in a manageable range to avoid numerical instability in the NCDE trajectories.

## D    Batching Irregular Time-Series

In Section 3 we introduced the pseudo-time method for training continuous models using batches. This relies on the equality

$$\text{ODESolve}(f, \boldsymbol{h}_0, t_0, t) = \text{ODESolve}(\tilde{f}, \boldsymbol{h}_0, s_0, s).$$

where $\tilde{f}(\boldsymbol{h}, s) = f(\boldsymbol{h}, \psi(s))\frac{d\psi}{ds}$ and that we interpolate $t$ with an increasing, surjective, piece-wise differentiable function so that $t = \psi(s)$. We prove this below.

We carry out an ODE solve over an infinitesimal pseudo-time step from $s_0$ to $s_0 + \Delta s$. We have $t_0 = \psi(s_0)$, $t_0 + \Delta t = \psi(s_0 + \Delta s)$. For small enough $\Delta s$ we have $\Delta t = \frac{d\psi}{ds}\Delta s$. We solve the ODE for the initial condition $\tilde{\boldsymbol{h}}(s_0) = \boldsymbol{h}_0$, and $\frac{d\tilde{\boldsymbol{h}}}{ds} = \tilde{f}(\tilde{\boldsymbol{h}}, s)$. Carrying out the ODE solve over the infinitesimal step we have:

$$\tilde{\boldsymbol{h}}(s_0 + \Delta s) = \tilde{\boldsymbol{h}}(s_0) + \tilde{f}(\tilde{\boldsymbol{h}}(s_0), s_0)\Delta s$$
$$= \tilde{\boldsymbol{h}}(s_0) + f(\tilde{\boldsymbol{h}}(s_0), \psi(s_0))\frac{d\psi}{ds}\Delta s$$
$$= \boldsymbol{h}_0 + f(\boldsymbol{h}_0, t_0)\Delta t$$

We also carry out a second ODE solve from $t_0$ to $t_0 + \Delta t$, where the initial condition is $\boldsymbol{h}(t_0) = \boldsymbol{h}_0$ and $\frac{d\boldsymbol{h}}{dt} = f(\boldsymbol{h}, t)$. Carrying out the ODE solve over this infinitesimal step we have:

$$\boldsymbol{h}(t_0 + \Delta t) = \boldsymbol{h}_0 + f(\boldsymbol{h}_0, t_0)\Delta t$$

Giving us $\boldsymbol{h}(t_0 + \Delta t) = \tilde{\boldsymbol{h}}(s_0 + \Delta s)$. ODEs are fully determined by their dynamics and initial condition,[3] so we can repeat the above step making $\boldsymbol{h}(s_0 + \Delta s)$ the new initial condition. This can be repeated for the whole $s$ domain, completing the proof. We demonstrate the batching procedure pictorially in Figure 2.

From the proof we can see why we need $\psi$ to be increasing, surjective and piece-wise differentiable. We require in the limit that $\Delta t = \frac{d\psi}{ds}\Delta s$ to carry out the infinitesimal solve, since $\Delta t \geq 0$ and $\Delta s \geq 0$, we require $\frac{d\psi}{ds}$ to both exist (differentiable) and be $\geq 0$, so we need an increasing function. As well as this we need to be able to carry out this step anywhere on the $t$ interval so $\psi$ must be surjective. Finally, we do not need to be globally differentiable, since for a whole ODE solve we can split the interval into non-overlapping sub-intervals and solve the ODE sequentially on those. Therefore $\psi$ only needs to be piece-wise differentiable. Examples of interpolations that fulfill this criteria are the linear and mono cubic interpolations.

**Computational and Memory Requirements.**    We use pseudo-times to achieve batching as a simple solution that is software agnostic and reuses already calculated interpolations from the Neural CDE. However, there is at least some theoretical evidence to suggest it would be more memory efficient than the method of taking unions of time-series and more compute efficient specifically for Neural CDEs. This is also supported by literature.

On page 60 of (Kidger, 2021), Section 3.2.1.3, Kidger remarks while discussing the scheme of Morrill et al. (2021) for Neural CDEs: "Remark 3.12. This was actually a mistake we made in [Kid+20a] – the above

---

[3]The Picard-Lindelöf theorem requires that the dynamics function be Lipschitz continuous (Lindelöf, 1894). This is achieved with ReLU activations.

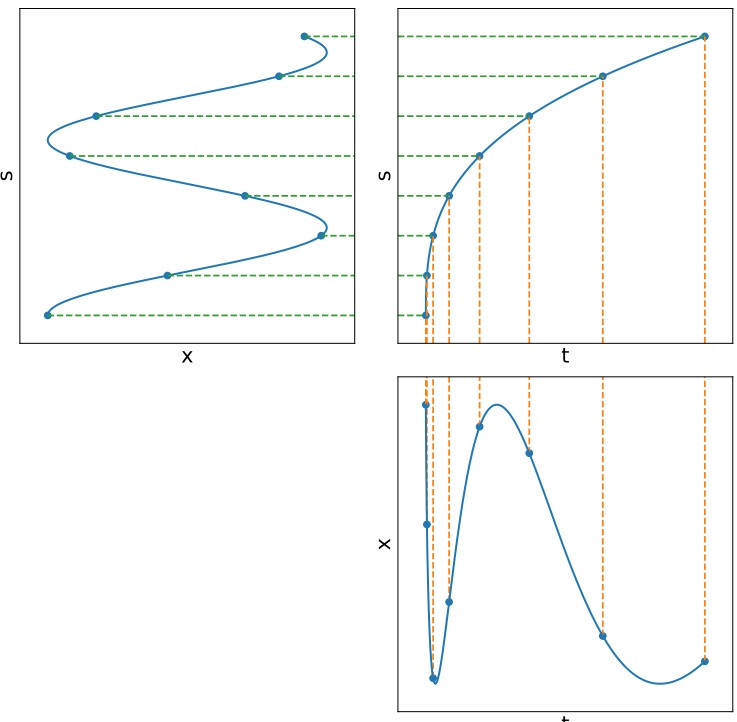

Figure 2: Visualization of the reparameterization from $s$ to $t$. In the top left is a regular time-series where $s$ represents pseudo-time (this panel is rotated 90° anti-clockwise). In the top right is how we map from $s$ to $t$ $(t = \psi(s))$, in the bottom right we have the irregular time-series where $t$ is true time. When we solve we include $\frac{d\psi}{ds}$ in the dynamics function, which is effectively using the top right figure as a way to transform from $s$ to $t$. Each time-series has a different version of this since the time-series are irregular, but these are all calculated before training so can be included easily for efficient batching.

procedure was not done, in favour of an alternate (storage-inefficient) scheme that involved taking the union over the times $t_j$ needed to handle each batch of data." That is, instead of the method that we propose (which is a generalization of (Morrill et al., 2021)), the times for each series in the batch are merged & sorted, then the combined system is solved for all times and only certain outputs are used in the loss calculation using a binary mask. So the existing literature supports this method over the union method of batching.

We can also show this by considering the complexities of the two methods. In the following, assume that all time-series in the dataset are length $n_t$, the analysis can be straightforwardly adapted when this is not the case. Assume that all initial times are the same, which is a requirement of merging the time-series to train in batches. We also assume that no two time-series share any evaluation times, if they do then these are combined when merging the sorted lists.

We start by considering the method of taking unions. First the union must be taken over sets of times in the batch and sorted. Then a binary mask is created to indicate whether an output is used in the loss or metric calculation. This has memory cost $\mathcal{O}(n_b^2 n_t)$ where $n_b$ is the batchsize, i.e. for each series in the batch (size $n_b$) we need a binary mask of length $n_b n_t$, the total number of merged times. Further to this, there is a further memory cost during the ODE solve, here we calculate the hidden state at $n_b n_t$ times, which must be stored (for loss calculation) for each hidden state in the batch giving a further memory requirement of $\mathcal{O}(n_b^2 n_t n_h)$, where $n_h$ is the size of the hidden state. Combining these two costs we have $\mathcal{O}(n_b^2 n_t (n_h + 1))$ memory cost.

Now we consider the computational cost. Assuming that during training there is shuffling of the series to create new batches each epoch, then we must merge and sort the time-series at every iteration. Merging the sorted lists is cost $\mathcal{O}(n_b n_t)$, there are $n_s/n_b$ iterations per epoch where $n_s$ is the size of the train set, so

merging and sorting has computational cost $\mathcal{O}(n_e n_s n_t)$ over the whole training, where $n_e$ is the number of epochs. For each solve we calculate the output of the ODE at $n_b n_t$ times, which means during the backward solve, we have to stop the solve explicitly this many times to adjust the adjoint state (which undergoes discrete jumps at measurement points, see (Chen et al., 2018) for an explanation of this). There is a cost of restarting an ODE solve (calculating the initial step size etc.) which if we treat as constant gives us $\mathcal{O}(n_b n_t)$ additional computational cost. This is done for every iteration of every epoch which gives a cost of $\mathcal{O}(n_e n_s n_t)$. So combining with the cost of sorting we have additional $\mathcal{O}(2n_e n_s n_t)$ to train.

Now we can compare to the pseudo-time method starting with the memory complexity. We need to now store cubic coefficients of the interpolation. This gives four coefficients for each point in each series giving memory complexity $\mathcal{O}(4n_s n_t)$. If we consider the memory requirement of the solve, we need to store the hidden state at all times during the solve which is reduced to $\mathcal{O}(n_b n_t n_h)$. Adding these two gives the total memory cost $\mathcal{O}(n_b^2 n_t(4n_s/n_b^2 + n_h/n_b))$. Comparing to the union method we directly compare $n_h + 1$ for the union method to $4n_s/n_b^2 + n_h/n_b$, where we have disregarded the common factor of $n_b^2 n_t$. Whilst this is dependent on specific values of $n_h$, $n_b$ and $n_s$, we can see that as the batchsize increases the pseudo-time method will become more memory efficient than the union method (as supported by (Kidger, 2021)).

Next we look at the computational complexity. As given in Section 5, to calculate the interpolation $\psi$, we have to interpolate each time-series individually which gives cost $\mathcal{O}(n_f n_s n_t)$ where $n_f$ is the number of features. This is a single preprocessing step so does not need to be done at each training iteration, so we do not consider it here. If we consider the computational cost during training, we only need to solve the ODE for $n_t$ times so the computational cost of restarting the ODE using the adjoint method for every iteration of every epoch is $\mathcal{O}(n_e n_s n_t/n_b)$. Finally we have to calculate $\psi(s)$ and $\frac{d\psi}{ds}$ at every function evaluation during the solve to calculate $\frac{d\boldsymbol{h}}{ds} = f(\boldsymbol{h}, \psi(s))\frac{d\psi}{ds}$. As described in Appendix E this involves taking a floor operation, and calculating a cubic. Which is constant, but must be done at each function evaluation. This is done for every sample in the batch (size $n_b$) at every iteration of training giving $\mathcal{O}(n_e n_s n_n)$ where $n_n$ is the number of dynamics function evaluations during the solve. Combining these we have cost $\mathcal{O}(n_e n_s n_t/n_b + n_e n_s n_n)$. Comparing this to the union method which had cost $\mathcal{O}(2n_e n_s n_t)$, we can see that the computational costs depend on $n_n$ vs $n_t$. Using advanced solvers $n_n$ is independent of $n_t$, so the computational cost is inconclusive and depends on $n_n$, i.e. how complex the hidden dynamics are. A significant exception is for Neural CDEs, the best performing model we benchmarked. Here the computational cost for the pseudo-time method is unchanged. However, for the union method the dynamics include $\frac{dX}{dt}$ (where the true time is used to interpolate the features). Now we include the computational cost associated with calculating $\frac{dX}{ds}$ or $\frac{dX}{dt}$ at every step. We also include the computational cost of calculating the interpolation coefficients. So now the union method involves all the computational costs that the pseudo-time method does, it also includes a further $\mathcal{O}(2n_e n_s n_t)$ cost associated with merging sets of sorted times and restarting the backward solve at more times. So theoretically we can see that for Neural CDEs the pseudo-time method is computationally more advantageous than the union method.

So we have theoretical evidence to suggest the pseudo-time method is more memory efficient than the union method, but the computational complexity is problem dependent (how many function evaluations are needed during the ODE solve), except for Neural CDEs where the pseudo-time method is theoretically faster. Again note that this has not been experimentally verified but is supported by Kidger (2021).

# E  Calculating Interpolations

In this section we describe how a general control signal is calculated without using any specific interpolation scheme and in Appendix F we discuss the interpolation schemes themselves. We use piece-wise cubics to interpolate, we don't use polynomials of arbitrary order since they can oscillate significantly.

Assuming we use integers as pseudo-time to interpolate, given a point $s$ to evaluate at, a control signal will first find the largest integer $i$ which is smaller than $s$ (floor function), and then $X(s)$ can be calculated using

$$X(s) = a_i(s - i)^3 + b_i(s - i)^2 + c_i(s - i) + d_i. \tag{3}$$

And its derivative can be calculated using

$$\frac{dX}{ds} = 3a_i(s-i)^2 + 2b_i(s-i) + c_i. \tag{4}$$

It is possible to expand these brackets, and then we can calculate new coefficients so that

$$X(s) = \tilde{a}_i s^3 + \tilde{b}_i s^2 + \tilde{c}_i s + \tilde{d}_i.$$

However, we found that this can be numerically unstable thus we use the method given by Equation 3. With this, the interpolation step finds the coefficients $a_i$, $b_i$, $c_i$ and $d_i$, which change depending on the interpolation scheme.

To deal with missingness, consider one feature of a partially observed time-series with $n$ measurements:

$$\big((x_0, 0), (\emptyset, 1), (x_2, 2), ..., (\emptyset, n-3), (x_{n-2}, n-2), (x_{n-1}, n-1)\big)$$

Here we are missing measurements of this feature at $s = 1$ and $s = n - 3$. So we remove these:

$$\big((x_0, 0), (x_2, 2), ..., (x_{n-2}, n-2), (x_{n-1}, n-1)\big)$$

We then run the interpolation on what is left getting $a$, $b$, $c$, $d$ coefficients for these observations. Finally to fill in the missing observations, we must continue the previous cubic piece. For example; between $s = 3$ and $s = 4$, we would like to continue the cubic that is between $s = 2$ and $s = 3$. To do this we require the two polynomials to match at all $s$:

$$a_i(s-i)^3 + b_i(s-i)^2 + c_i(s-i) + d_i = a_{i+1}(s-(i+1))^3 + b_{i+1}(s-(i+1))^2 + c_{i+1}(s-(i+1)) + d_{i+1}$$

Solving this, to find the coefficients where we have a missing value we take a linear combination of the previous coefficients

$$\begin{bmatrix} a_{i+1} \\ b_{i+1} \\ c_{i+1} \\ d_{i+1} \end{bmatrix} = \begin{bmatrix} 1 & 0 & 0 & 0 \\ 3 & 1 & 0 & 0 \\ 3 & 2 & 1 & 0 \\ 1 & 1 & 1 & 1 \end{bmatrix} \begin{bmatrix} a_i \\ b_i \\ c_i \\ d_i \end{bmatrix}$$

For $i = -1$ or $i = n$, i.e. outside the range of the time-series, we set $a = b = c = 0$ and $d_{-1} = d_0$ or $d_n = d_{n-1}$. That is, outside the range of the time-series the control signal predicts either the first or the last observed values as a constant, with zero derivative.

Note, since we have no knowledge about the missingness of features, or the irregularity of the time-series we must carry out this procedure separately for each feature of each time-series. This is a slow process and therefore is part of data preprocessing and the coefficients are stored.

## F  Interpolation Schemes

Here we give information about the specific interpolation schemes used, including the positives and negatives of using each. The majority of this is summarised from (Morrill et al., 2021). Importantly, for the $t$ channel we must use a control signal that is increasing, surjective and piece-wise differentiable. We would also like to use an increasing interpolation for the observation counts since these can only increase and we can enforce this inductive bias. In the following we give the steps for finding the coefficients to interpolate, this assumes a general set of times, rather than the integer pseudo-times we use. Therefore these can be used generally, and the specific case of the integers is a trivial change.

### F.1  Linear

This is the simplest interpolation scheme, we carry out a linear interpolation between two observed points, so

$$\begin{bmatrix} a_i \\ b_i \\ c_i \\ d_i \end{bmatrix} = \begin{bmatrix} 0 \\ 0 \\ (x_{i+1} - x_i)/(t_{i+1} - t_i) \\ x_i \end{bmatrix}$$

**Pros and Cons.** This is the simplest interpolation scheme, and intuitively the easiest to understand. If the observations are increasing then the control signal is an increasing function, therefore this can be used to interpolate time and observation counts. However, it does not have a continuous first derivative so is slow to use in an ODE solver, it also misrepresents reality, it is unlikely features change in such a way that the derivative is discontinuous at points.

## F.2 Hermite Cubic

Hermite interpolation is a smooth version of linear interpolation, where rather than discontinuities in the derivative at observations these are smoothed by making the derivative continuous there. Let the derivative at each point be equal to the derivative that would be calculated by linearly interpolating with the previous observation, $m_i = (x_i - x_{i-1})/(t_i - t_{i-1})$ for $i \geq 1$. Then set $m_0 = m_1$. With this we can calculate the coefficients

$$\begin{bmatrix} a_i \\ b_i \\ c_i \\ d_i \end{bmatrix} = \begin{bmatrix} (-2(x_{i+1} - x_i) + (t_{i+1} - t_i)(m_i - m_{i+1}))/(t_{i+1} - t_i)^3 \\ (3(x_{i+1} - x_i) - (t_{i+1} - t_i)(2m_i - m_{i+1}))/(t_{i+1} - t_i)^2 \\ m_i \\ x_i \end{bmatrix}$$

**Pros and Cons.** This is a smoothed out version of the linear interpolation, so is faster to use in a solve. Additionally, as it is smooth it is a better inductive bias for interpolating features. However, it cannot be used to interpolate time or observation count since it is not increasing. Also whilst having a continuous first derivative, it does not have a continuous second derivative.

## F.3 Monotonic Cubic

We desire a scheme with continuous first derivative to interpolate time and observation count. However, Hermite cubics are not increasing functions and piece-wise linear does not have continuous derivatives. To overcome this we propose a new interpolation scheme, the monotonic cubic (mono cubic). This is a Hermite cubic but we enforce zero derivative at every observation, therefore it is an increasing function if the observations are always increasing. It has continuous first derivative so we can use it in place of linear and it is increasing and surjective so we can use it in place of the Hermite interpolation.

The formula for the coefficients is found by taking the formula for the Hermite coefficients and setting $m_i = 0$ for all $i$.

$$\begin{bmatrix} a_i \\ b_i \\ c_i \\ d_i \end{bmatrix} = \begin{bmatrix} -2(x_{i+1} - x_i)/(t_{i+1} - t_i)^3 \\ 3(x_{i+1} - x_i)/(t_{i+1} - t_i)^2 \\ 0 \\ x_i \end{bmatrix}$$

**Pros and Cons.** The mono cubic is both smoother than the linear interpolation and is increasing unlike the Hermite cubic. Therefore it is good to use for interpolating time or observations counts. However, it is not as good for interpolating features, since zero derivative is enforced at the observation points which is not a realistic inductive bias.

## F.4 Natural Cubic

The natural cubic is a piece-wise cubic that is twice-continuously differentiable everywhere. So the function, its first derivative and its second derivative are all continuous at and between observation points.

We start by constructing a system of linear equations for a vector $k$

$$k_{i-1}(t_{i-1} - t_i) + 2k_i(t_{i-1} - t_{i+1}) + k_{i+1}(t_i - t_{i+1}) = 6\left( \frac{x_{i-1} - x_i}{t_{i-1} - t_i} - \frac{x_i - x_{i+1}}{t_i - t_{i+1}} \right)$$

This is for $0 < i < n - 1$. We also have boundary conditions $k_0 = 0$ and $k_n = 0$. This system of linear equations can be solved using the tridiagonal algorithm which is $\mathcal{O}(n)$, rather than $\mathcal{O}(n^3)$ for Gaussian

elimination. After finding the vector $k$, its elements can be used to find the coefficients.

$$\begin{bmatrix} a_i \\ b_i \\ c_i \\ d_i \end{bmatrix} = \begin{bmatrix} (k_{i+1} - k_i)/6(t_{i+1} - t_i)^3 \\ k_i/2 \\ (x_{i+1} - x_i)/(t_{i+1} - t_i) - (k_{i+1} + 2k_i)(t_{i+1} - t_i) \\ x_i \end{bmatrix}$$

From the above formula we see that the second derivative of the interpolation at point $i$ is $k_i$. Additionally, from the formula for $c_i$ we see that the derivative at point $i$ is what we expect from linear control signals $(x_{i+1} - x_i)/(t_{i+1} - t_i)$ with a corrective term to make the first and second derivatives continuous.

**Pros and Cons.** This is the smoothest of the interpolants with a continuous second derivative. This makes the ODE solve faster. However, it is not physical since it uses the whole series to find the coefficients as part of the tridiagonal solve. This means natural cubic can only be used when we enforce physical solutions using the method of copying subjects, this cannot be used with recti control signals.

## G  Library Specific Caveats

Our implementation uses TensorFlow and Keras, for which a differentiable ODE Solver exists. This can be found at `https://www.tensorflow.org/probability/api_docs/python/tfp/math/ode`. We found two main caveats using this that do not necessarily exist in other libraries such as TorchDiffEq (PyTorch) (Chen et al., 2018) or Diffrax (JAX) (Kidger, 2022). We provide the straightforward fixes here to aid implementation.

### G.1  Running an ODE Backwards in Time

For models such as the Latent ODE, an ODE is solved backwards in time, this is when we solve from $t_1$ to $t_0$ where $t_1 \geq t_0$. The TensorFlow solver does not handle this by design, it assumes $t_1 \leq t_0$ (time is ordered). We are able to overcome this by solving a related ODE, with a new ODE function. If previously

$$\frac{d\boldsymbol{h}}{dt} = f(\boldsymbol{h}, t)$$

we use a new ODE

$$\frac{d\boldsymbol{h}}{dt} = \tilde{f}(\boldsymbol{h}, t) = -f(\boldsymbol{h}, -t).$$

And now we solve from $-t_1$ to $-t_0$ with the new ODE function

$$\boldsymbol{h}(t_0) = \text{ODESolve}(\tilde{f}, \boldsymbol{h}(t_1), -t_1, -t_0).$$

And since $-t_1 \leq -t_0$, the Tensorflow ODE solver can solve this. This requires that

$$\text{ODESolve}(\tilde{f}, \boldsymbol{h}(t_1), -t_1, -t_0) = \text{ODESolve}(f, \boldsymbol{h}(t_1), t_1, t_0)$$

which we prove below.

Let $\tilde{t} = -t$, and solve a new ODE for $\tilde{\boldsymbol{h}}(\tilde{t})$, where $\tilde{\boldsymbol{h}}(\tilde{t}_1) = \boldsymbol{h}(t_1)$. We carry out a solve over an infinitesimal step from $\tilde{t}_1$ to $\tilde{t}_1 + \Delta\tilde{t}$ giving

$$\begin{aligned} \tilde{\boldsymbol{h}}(\tilde{t}_1 + \Delta\tilde{t}) &= \tilde{\boldsymbol{h}}(\tilde{t}_1) + \tilde{f}(\tilde{\boldsymbol{h}}(\tilde{t}_1), \tilde{t}_1)\Delta\tilde{t} \\ &= \tilde{\boldsymbol{h}}(\tilde{t}_1) - f(\tilde{\boldsymbol{h}}(\tilde{t}_1), -\tilde{t}_1)\Delta\tilde{t} \\ &= \boldsymbol{h}(t1) + f(\boldsymbol{h}(t_1), t_1)\Delta t. \end{aligned}$$

Which is equal by definition to $\boldsymbol{h}(t_1 + \Delta t)$, (where $\Delta t$ is negative). As with the proof of minibatching in Appendix D, repeatedly applying this gives $\boldsymbol{h}(t_0) = \tilde{\boldsymbol{h}}(\tilde{t}_0)$, completing the proof. So we can solve an ODE backwards by solving a related ODE forwards with initial and final times multiplied by -1.

### G.2 Repeated Calls to an ODE Solver

The TensorFlow ODE solver is able to take a sorted list of times to solve up to, and it functions fully in this scenario. Therefore for models like Neural CDE and GRU-ODE we do not need to repeatedly call the solver, we can just ask it to solve for all times of interest. For the ODE-RNN and by extension Latent ODE (as it uses an ODE-RNN encoder), we must repeatedly call the solver. This is because we discontinuously change the state at observations, and so we must stop the ODE solve, update the hidden state and call the solve again.

Repeatedly calling the solver prevents the model from backpropagating, preventing the models from training. In fact it doesn't simply prevent the models from training, but the program stops running. The fix to this problem is to explicitly pass parameters into the ODE function, and then also pass these as "constants" to the ODE solver, so that it knows to calculate their gradients. We include simple example code below:

This code firstly defines the ODE function that explicitly has parameters written out and named. We demonstrate a two hidden layer ReLU multilayer perceptron with tanh final non-linearity (as is the case with our dynamics functions). Following that we show how it is used with a TensorFlow solver.

```
import tensorflow as tf
import tensorflow_probability as tfp

class RepeatableODEFunc(tf.keras.layers.Layer):

    def __init__(self, x_dim, h_dim):
        super().__init__()
        self.w1 = tf.Variable(tf.random.normal(shape=(x_dim, h_dim)), trainable=True)
        self.b1 = tf.Variable(tf.random.normal(shape=(h_dim,)), trainable=True)
        self.w2 = tf.Variable(tf.random.normal(shape=(h_dim, h_dim)), trainable=True)
        self.b2 = tf.Variable(tf.random.normal(shape=(h_dim,)), trainable=True)
        self.w3 = tf.Variable(tf.random.normal(shape=(h_dim, x_dim)), trainable=True)
        self.b3 = tf.Variable(tf.random.normal(shape=(x_dim,)), trainable=True)

    def call(self, t, x, w1, b1, w2, b2, w3, b3):
        x = tf.nn.relu(tf.matmul(x, w1) + b1)
        x = tf.nn.relu(tf.matmul(x, w2) + b2)
        x = tf.math.tanh(tf.matmul(x, w3) + b3)
        return x

ode_fn = RepeatableODEFunc(x_dim, h_dim)
x_next = tfp.math.ode.DormandPrince().solve(
    ode_fn,
    t_now,
    x_now,
    solution_times=[t_next],
    constants = {
        "w1": ode_fn.w1,
        "b1": ode_fn.b1,
        "w2": ode_fn.w2,
        "b2": ode_fn.b2,
        "w3": ode_fn.w3,
        "b3": ode_fn.b3,
    },
).states[-1]
```

