# OpenReview forum: "Benchmarking Continuous Time Models for Predicting Multiple Sclerosis Progression"
_TMLR — Accepted by TMLR_

### Review · Reviewer_eSNb · 2023-06-09

**Summary Of Contributions:**

This paper presents the use of continuous time models, particularly neural C/ODEs, for longitudinal modeling of multiple sclerosis. This is an application-oriented studies with some methodological innovations in interpolation schemes and mini-batching strategies.Experiments were conducted to compare CDE and a variety of neural ODE variants to temporal convolutional networks (TCNs).

**Audience:**

Yes

**Broader Impact Concerns:**

No concern.

**Claims And Evidence:**

No

**Requested Changes:**

Please see the detailed list above for a) necessary addition of experiments in ablation and baselines (1/4/6), 2) addition of clarification with respect to contribution of works (1/2/3/5), and 3) further investigations on key aspects of the experimental works (7/8/9)

**Strengths And Weaknesses:**

The presented work appear to be overall focused on application of CDEs, whereas the technical innovations and contributions are incremental and limited. For an application-focused paper, the empirical results are also relatively limited to support the stated contributions.

1. The two main technical points, interpolate and mini-batching strategies, were motivated to have memory and computational gains. These however were not supported or ablated in experiments. Additionally, the computational requirements needed to pre-process the splines over the datasets should be discussed (which typically can be costly).

2. The ablation results do not show significant difference between the interpolation schemes, which make it difficult to assess the contribution of this technical point.

3. Parallel-time solvers that decouple the batch already exist. (torchode https://openreview.net/pdf?id=uiKVKTiUYB0). Relation between the presented pseudo-timing training and these related works should be discussed.

4. The baseline models included, both in terms of discrete time models and continuous time models, are limited.

5. Other than CDE, the rest of the NODE models do not show any improvements over the TCNs. This contradicts the motivation of the work about the advantage of continuous time models.

6. Two schemes which enforce the models to only look at past data are introduced, but there is no data to show how they differ in terms of performance.

7. The reduced feature set outperforming all features warrants further investigations and reasoning.

8. Standardization works better appears to be one of the takeaways. The author posit it is because standardization takes X to a manageable range, however ddid not extend on it further. The effect of interpolation upon standardization also seems nominal, which the authors have pointed out. More insights on why could be helpful

9. 	More details on ψ can be helpful. How is ψ determined? Its nature? Requirements? Will any arbitrary increasing function work? To me, it does not feel like it will? So, which functions work? Does the choice of function affect modelling in any way?




Minor comments.
	•	Using c to represent both context vector and spline coefficients is confusing at times.
	•	The word “causal” is used in a sense that it might not necessarily imply causation or the effect, as is common in literature surrounding do-calculus and causal statistics.

---

> ### Author Response · Authors · 2023-06-30
> **Official Response to Reviewer eSNb**
>
> Firstly we would like to thank you for taking the time and effort to write a very detailed and constructive review. We are happy to implement many of the suggested changes and are confident it will improve the paper. The concerns raised are summarised below:
>
> 1. Use of $\psi$ to batch and computational requirements
> 2. Interpolation schemes showing similar results
> 3. Comparison to torchode
> 4. Number of baselines
> 5. Only CDE improving over TCN
> 6. Comparisons between two preprocessing schemes for physical solutions
> 7. Reduced Feature Set
> 8. Standardization of $X$
> 9. Further details about $\psi$
> 10. Minor Comments
>
> We will happily answer these points in separate replies to keep discussion distinct. We will also update the manuscript and inform all reviewers when this is uploaded.

---

> ### Author Response · Authors · 2023-06-30
> **1. The use of $\psi$ to batch and computational requirements**
>
> We do not believe that ablations are necessary in this case, since the speed and memory requirements are supported both by existing literature and the asymptotic complexities of the methods.
>
> On page 60 of Kidger 2022, Section 3.2.1.3, Kidger remarks while discussing a related scheme for Neural CDEs: “Remark 3.12. This was actually a mistake we made in [Kid+20a] – the above procedure was not done, in favour of an alternate (storage-inefficient) scheme that involved taking the union over the times $t_j$ needed to handle each batch of data.” That is, instead of the method that we propose (which is a generalisation from Morrill et al. 2021), the times for each series in the batch are merged & sorted, then the combined system is solved for all times and only certain outputs are used in the loss calculation using a binary mask. So the existing literature supports this method over the union method of batching.
>
> We can also show this by considering the complexities of the two methods. In the following, assume that all time-series in the dataset are length $n_t$, the analysis can be straightforwardly adapted when this is not the case. Assume that all initial times are the same, which is a requirement of merging the time-series to train in batches (our method does not have this requirement). We also assume that no two time-series share any evaluation times, if they do then these are combined when merging the sorted lists, the complexity is still the same.
>
> We start by considering the method of taking unions. First the union must be taken over sets of times in the batch and sorted. To merge the sorted lists is $\mathcal{O}(n_b n_t)$ where $n_b$ is the batchsize. As well as this there is a memory cost of $\mathcal{O}(n_b^2 n_t)$ which is a binary mask to indicate whether a result at a given time for a given series in the batch is relevant for the loss calculation, i.e. for each series in the batch we need a binary mask of length $n_b n_t$, the total number of merged times. During the solve we calculate the output of the ODE at $n_b n_t$ times, which means during the backward solve we have to stop the solve explicitly this many times to adjust the adjoint state (which undergoes discontinuous jumps at measurement points, see Chen et al. 2018 for an explanation of this). This slows down the backward solve significantly since there is a cost of restarting an ODE solve. Assuming that during training there is shuffling of the series to create new batches each epoch, then this must be done for every iteration of training.
>
> In contrast, the method of solving the ODE in a related domain leads to the following speed ups. To calculate the interpolation $\psi$, we have to interpolate each time-series individually which gives cost $\mathcal{O}(n_s n_t)$ where $n_s$ is the total number of subjects. Note $n_s$ is the sum of all the batch sizes from a single epoch, however this is a single preprocessing step so does not need to be done at each training iteration, therefore assuming we train for more than one epoch, interpolating is faster. Regarding the memory complexity, we need to now store cubic coefficients of the interpolation. This gives four coefficients for each point in each series giving memory complexity $\mathcal{O}(4n_s n_t)$. The memory complexity of storing the binary mask in the alternative method was $\mathcal{O}(n_b^2 n_t)$ which, depending on batch size, will often be more memory inefficient (for a train set of size 10,000 a batch size of 200 is less efficient), as noted in Kidger 2022 above, this is often the case. Finally during the solve we only need to solve for $n_t$ times, which during the backwards solve of the adjoint method provides significant speedup. We will add this analysis and the citation of Kidger 2022 to the explanation of our choice to batch in this way.
>
> Regarding the computational cost of interpolating the whole dataset, indeed this can be costly. We note at the start of Section 5 this is $\mathcal{O}(n_f n_t n_s)$, where $n_s$ is the number of time-series in the dataset, $n_t$ is the average length of those time-series and $n_f$ is the number of features. This is because each feature must be treated separately. We mention in Section 3.1 that for this reason we calculate the interpolations once and store the coefficients. Additionally this process can be parallelized if speed is a particular concern which we list as one of our future directions of work.
>
> # References
>
> 1. Kidger, P., 2022. On neural differential equations. arXiv preprint arXiv:2202.02435.
> 2. Morrill, J., Kidger, P., Yang, L. and Lyons, T., 2021. Neural controlled differential equations for online prediction tasks. arXiv preprint arXiv:2106.11028.
> 3. Chen, R.T., Rubanova, Y., Bettencourt, J. and Duvenaud, D.K., 2018. Neural ordinary differential equations. Advances in neural information processing systems, 31.

---

> ### Author Response · Authors · 2023-06-30
> **2. Interpolation Schemes show similar results**
>
> Indeed there is no significant difference between the interpolation schemes. This is because the learnable part of the dynamics function (the $f$ matrix in $dz/ds = f(z)\frac{dX}{ds}$ is able to adjust for the different interpolation schemes to achieve the best results. As we say in the paper, this is consistent with Morrill et al. 2021 where they have introduced multiple interpolation schemes (in particular the rectilinear scheme) as technical contributions and found that they all perform effectively the same. And so it is good that our results are consistent with previously found results on different datasets. We also would like to respectfully add that we do not believe only positive results make a contribution valid. We have introduced two further interpolation schemes as additional options when modelling with Neural CDEs, which perform to the same standard as the pre-existing schemes, we do not claim that they are necessarily better than the existing schemes.
>
> We also accept that our contributions section has made the paper confusing. Our main contribution is testing continuous time models on a real Multiple Sclerosis dataset to show that if a continuous model is well tuned it can outperform discrete time models, paving the way for further research. The new interpolation and batching methods are contributions but we do not want that to dilute the main message so shall rewrite our introduction to make the point that these are secondary contributions and the benchmarking offour continuous models is the primary contribution.
>
> # References
>
> 1. Morrill, J., Kidger, P., Yang, L. and Lyons, T., 2021. Neural controlled differential equations for online prediction tasks. arXiv preprint arXiv:2106.11028.

---

> ### Author Response · Authors · 2023-06-30
> **3. Comparison to torchode**
>
> torchode (Lienen and Günnemann 2022) is a relatively new work that speeds up Neural ODEs relative to the torchdiffeq package by improving the low level implementation of the solvers. This is done by treating each ODE in a batch separately with its own solver state and integration bounds, and solving each ODE separately in parallel. In contrast, our method is a higher level method that does not require a new package or low level implementation. We use a continuous, increasing piecewise differentiable interpolation of $t$ given by $t = \psi(s)$ where $s$ are the integers. This allows batch training using libraries that do not treat the batches independently, such as the TensorFlow solver we used https://www.tensorflow.org/probability/api_docs/python/tfp/math/ode/Solver. The paper torchode itself notes some possible shortcomings of their implementation:
>
> “The extra cost incurred by tracking a separate solver state for every problem is negligible on a highly parallel computing device such as a GPU. However, because each ODE progresses at a different pace, they might pass a different number of evaluation points at each step. Keeping track of this requires indexing with a Boolean tensor, a relatively expensive operation.”
>
> That is, certain parallel hardware is required (GPUs), which is rarely an issue but could be for certain low resource applications or researchers. Diffrax (Kidger 2022) is the only other existing library that is able to batch over different regions of integration, which provides further capabilities, but is similarly a low level implementation and achieves parallelism in the same way. We shall discuss both of these in the related work.
>
> Finally, as noted in our previous reply, while listed as a contribution, this was not the major focus of the work. The main message was about using continuous methods to model Multiple Sclerosis development, and we showed that using the right continuous model can outperform the previous best tested discrete-time model. We shall rephrase the introduction to make this distinction clear. Both the interpolations to fill missing values and the interpolation of $\psi$ were already calculated as necessary steps to construct the control signal $X(s)$ for the Neural CDE. The method of using the interpolations to parallelize the batches was used as a high level solution using existing interpolations, rather than converting code to a different library or implementing a low-level parallel solver.
>
> # References
>
> 1. Lienen, M. and Günnemann, S., 2022. torchode: A Parallel ODE Solver for PyTorch. arXiv preprint arXiv:2210.12375.
> 2. Kidger, P., 2022. On neural differential equations. arXiv preprint arXiv:2202.02435.

---

> ### Author Response · Authors · 2023-06-30
> **4. Number of Baselines**
>
> We believe that four continuous baselines is a sufficient number to test. Further to this we are comparing against the best performing model from Roy et al. 2022 so we are comparing against all models from that paper which is six discrete-time baselines covering a wide range of possible models. There is a model from each major category of techniques: logistic and linear models representing classical models; gradient boosted classifiers/regressors representing tree based methods; MLP as a simple deep learning approach and TCN as a deep learning approach for sequences. We do not include all of these in the tables since the results are already known and we only need to compare to the best model.
>
> # References
>
> 1. Roy, S., Mincu, D., Proleev, L., Rostamzadeh, N., Ghate, C., Harris, N., Chen, C., Schrouff, J., Tomašev, N., Hartsell, F.L. and Heller, K., 2022, April. Disability prediction in multiple sclerosis using performance outcome measures and demographic data. In Conference on Health, Inference, and Learning (pp. 375-396). PMLR.

---

> ### Author Response · Authors · 2023-06-30
> **5. Only CDE Improves over TCN**
>
> Whilst the three other models (ODE-RNN, Latent ODE, GRU-ODE) cannot beat TCN consistently, they are able to beat other models. If we compare to Table 2 in Roy et al. 2022 (which benchmarks the other models), we see for the EDSS Mean task, that GRU ODE performs poorly (because the dynamics are heavily restricted) but ODE-RNN and Latent ODE outperform every other model. For EDSS > 3 and EDSS > 5 tasks ODE-RNN, Latent ODE and GRU-ODE are able to outperform all other models again just not TCN. So they aren’t necessarily poor, they just aren’t as good as the best discrete-time model tested. We also believe that Neural CDE being the only model to consistently beat TCN is able to provide insight that is just as valuable as continuous models easily beating discrete models. That is that being continuous alone is not enough to beat a well-tuned discrete baseline and that good hyperparameter tuning and model selection must still take place. The results support the findings from Kidger et al. 2020, that Neural CDE is able to beat most continuous models due to its superior expressive power. So whilst the initial motivation for using continuous time models was not as effective as originally thought we still gain insight into the models.
> We mention this in Section 5.1 and Section 6 as a limitation, we shall rewrite this to make it more clear that being continuous does not guarantee performance gains.
>
> Analogously (but certainly not identically), increased depth in Neural Networks does not lead to performance gains (due to vanishing/exploding gradients), despite theoretically being able to have more complex representations in later layers and having more parameters. Architectural choices must be made to unlock the full potential such as skip connections in Residual Networks (He et al. 2016).
>
>
> # References
>
> 1. Roy, S., Mincu, D., Proleev, L., Rostamzadeh, N., Ghate, C., Harris, N., Chen, C., Schrouff, J., Tomašev, N., Hartsell, F.L. and Heller, K., 2022, April. Disability prediction in multiple sclerosis using performance outcome measures and demographic data. In Conference on Health, Inference, and Learning (pp. 375-396). PMLR.
> 2. Kidger, P., Morrill, J., Foster, J. and Lyons, T., 2020. Neural controlled differential equations for irregular time series. Advances in Neural Information Processing Systems, 33, pp.6696-6707.
> 3. He, K., Zhang, X., Ren, S. and Sun, J., 2016. Deep residual learning for image recognition. In Proceedings of the IEEE conference on computer vision and pattern recognition (pp. 770-778).

---

> ### Author Response · Authors · 2023-06-30
> **6. Comparisons between two schemes enforcing models to use the past only**
>
> We apologize that this is unclear. The two methods are used before interpolation to construct a dataset that is then interpolated, the two methods then use two different non-overlapping partitions of the interpolations provided as part of the preprocessing. The first method of creating multiple subjects to enforce physical behaviour encompasses the interpolation schemes Linear, Hermite Cubic, Natural Cubic and Mono Cubic. The second method of extending each subject longitudinally is RectiLinear and RectiCubic, we use the name recti since this was coined initially in Morrill et al. 2021 where the method of extending longitudinally to have physical solutions was introduced. This information is included in the description of each interpolation scheme in Section 3.3 but we shall make this more clear (we've already renamed the Causal Interpolation Paragraph to Causal Interpolation (Recti Method)). Therefore, in Table 3 the Linear, Hermite Cubic, Natural Cubic and Mono Cubic columns test the first method, and the RectiLinear and RectiCubic test the scond method. As mentioned by the reviewer, the difference is minimal between the two methods since the learnable function $f$ makes up for the different schemes, which is consistent with the findings in Morrill et al. 2021.
>
> Importantly, both methods enforce solutions to only use the past to predict the future, which is a vital requirement.
>
> # References
>
> 1. Morrill, J., Kidger, P., Yang, L. and Lyons, T., 2021. Neural controlled differential equations for online prediction tasks. arXiv preprint arXiv:2106.11028.

---

> ### Author Response · Authors · 2023-06-30
> **7. Reduced Feature Set**
>
> At the beginning of Section 5 we describe the subset of features used and why. To summarize, as given in Reply 1, the computational cost of interpolation is $\mathcal{O}(n_f)$ so the computational cost to interpolate increases linearly with the number of features. Additionally, for every call of the Neural CDE dynamics function there is a matrix-vector multiplication of cost $\mathcal{O}(n_f )$, which leads to increased computation not just during the interpolation or embedding the initial condition, but at every stage of the Neural CDE solve. The features chosen have the lowest attrition compared to other features, they are approximately 10 times as abundant at the start of the study increasing to 1000 times at 9 months compared to other features (see Figure 2 of Roy et al. (2022) for a chart tracking the feature sparsity over time).
>
> Moreover, we have run a feature ablation in Appendix B.3 where we compare the full feature set against the reduced set and see that the reduced set outperforms the full set every time. Included in Appendix B.3 is our discussion why this might be the case: “We further hypothesize that one of the reasons NCDE becomes unstable to a large number of features is because the dynamics function is given by a matrix-vector multiplication. The dynamics of the NCDE is given by $\frac{dh}{ds} = f_\theta^c(h, \psi(s))\frac{dX}{ds}$ which is a matrix-vector multiplication, whereas the other three models have a dynamics function given by $\frac{dh}{ds} = f_\theta^c(h, \psi(s))\frac{d\psi}{ds}$ which is a vector-scalar multiplication. This would mean, since all functions $f_\theta^c$ have a final tanh activation, each component of the NCDE dynamics are bounded by $\pm \sum_i \frac{dX_i}{ds}$, whereas each component of the other models dynamics are bounded by $\pm\frac{d\psi}{ds}$. This explains why NCDE is unstable to a large number of features, because the magnitude of the velocity of the hidden state grows with the number of features, this does not happen for the other three models. One possible solution would be to divide the NCDE dynamics by the dimensionality of the vector $X$, which would make the dynamics agnostic to the number of features. On the other hand, this could also be a further reason why NCDE performs better than the other models (as well as NCDE having provably more expressive power (Kidger et al. 2020)), the restriction on the dynamics is not as tight for NCDE models. It further explains the importance of standardizing $X$, keeping $\frac{dX}{ds}$ in a manageable range to avoid numerical instability in the NCDE trajectories.” We shall reference this part of the Appendix more clearly in the main paper.
>
> # References
>
> 1. Roy, S., Mincu, D., Proleev, L., Rostamzadeh, N., Ghate, C., Harris, N., Chen, C., Schrouff, J., Tomašev, N., Hartsell, F.L. and Heller, K., 2022, April. Disability prediction in multiple sclerosis using performance outcome measures and demographic data. In Conference on Health, Inference, and Learning (pp. 375-396). PMLR.
> 2. Kidger, P., Morrill, J., Foster, J. and Lyons, T., 2020. Neural controlled differential equations for irregular time series. Advances in Neural Information Processing Systems, 33, pp.6696-6707.

---

> ### Author Response · Authors · 2023-06-30
> **8. Standardization of $X$**
>
> In Appendix B.1 we run the same ablation as Table 3 on ODE-RNN, Latent ODE and GRU-ODE in Tables 9-11. The results of these ablations support our hypothesis, since standardization does not affect these models as much as Neural CDE, and they do not have $dX/ds$ in their dynamics. We write: “We see that on the whole ODE-RNN does best when standardizing features and filling missing values with zeros, Latent ODE does not have a clear superior scheme and GRU-ODE does best when standardizing features and filling missing values with interpolated values. However for these three methods the differences are small, whereas with Neural CDEs there was a clear distinction between standardizing and not standardizing features. This is likely due to the models’ dynamics functions. The dynamics function for the Neural CDE ends with $dX/ds$, whereas all others end with $d\psi/ds$. The true times are relatively small and therefore so is $d\psi/ds$. However, $X$ can be quite large, and therefore so is $dX/ds$. And so standardizing $X$ aids Neural CDEs more because $dX/ds$ in the dynamics is explicitly shifted to a more manageable range, whereas $d\psi/ds$ was already in that range”. This is further supported by Appendix B.3 where we run the feature ablation as discussed in Reply 7. Here we additionally hypothesize that since the Neural CDE dynamics function is a matrix-vector product rather than vector-scalar product for the other three models, and that the learnable part has a final tanh non-linearity each component of the Neural CDE dynamics are bounded by $\pm \sum_i \frac{dX_i}{ds}$. Whereas each component of the other three models’ dynamics are bounded by $\pm\frac{d\psi}{ds}$. This further supports the need to standardize the features. We shall add a sentence to the main paper to explain these results and point to them in the Appendix.
>
> As already mentioned we see that the effect of which interpolation scheme used is minimal, which is consistent with Morrill et al. 2021, since the learnable part of the Neural CDE dynamics function can account for the different interpolation schemes. We also see that filling missing values with interpolated values or zero does not have a significant effect, likely because we chose the features with the lowest attrition rates so there are not as many missing measurements as for the other features, which further supports our use of the reduced feature set.
>
> # References
>
> 1. Morrill, J., Kidger, P., Yang, L. and Lyons, T., 2021. Neural controlled differential equations for online prediction tasks. arXiv preprint arXiv:2106.11028.

---

> ### Author Response · Authors · 2023-06-30
> **9. Further details on $\psi$**
>
> We apologize that this was not made more clear in the paper. The requirements are the same as those in Morrill et al. 2021, if $\psi: [a, b] \xrightarrow{} [c, d]$, where $s \in [a, b]$ and $t \in [c, d]$ then $\psi$ must be increasing, differentiable and surjective on the interval $[a, b]$. From our proof in Appendix C, it is relatively straightforward to see why this must be the case. We require $\Delta t = \frac{d\psi}{ds} \Delta s$, so that we can solve the ODE over infinitesimal steps (in order to solve the ODE over a long range we must be able to also solve it over a small range). Since $\Delta t$ and $\Delta s$ are both $\geq 0 $, we require that $\frac{d\psi}{ds}$ both exists (requires differentiability) and is greater than or equal to zero (increasing function). If we are able to solve the ODE in the range $[a, b]$ which is $[c, d]$ in the $t$ domain, then we must also be able to solve the ODE for any sub-interval, which requires all times in $[c, d]$ to be mapped to, giving us surjectivity. We can simplify these constraints slightly, and split the $[a, b]$ and $[c, d]$ domains into non-overlapping sub-intervals - we can solve the ODE on the $[a, b]$ and $[c, d]$ domains by solving it sequentially on sub-intervals. Therefore the requirements are that $\psi$ is piece-wise differentiable, globally increasing and globally surjective. We describe the two interpolation methods we use for this in Section 3.3 and Appendix E, which are linear interpolation and our monocubic interpolation. However, we appreciate this is not clear and shall update the wording to be improve clarity.
>
> We see from Tables 3 and 12-14, which test the different interpolation schemes, that the choice of interpolation scheme does not significantly affect the performance. The linear interpolation of $\psi$ is used in the linear and rectilinear splines, all others use the monocubic interpolation of $\psi$. Which we have previously explained is due to the learnable part of the dynamics making up for these differences to achieve the best performance.
>
> # References
>
> 1. Morrill, J., Kidger, P., Yang, L. and Lyons, T., 2021. Neural controlled differential equations for online prediction tasks. arXiv preprint arXiv:2106.11028.

---

> ### Author Response · Authors · 2023-06-30
> **10. Minor Comments**
>
> - Context vs coefficients: Please could you point us to where we use $c$ to reference coefficients, as far as we can tell we use $c$ to refer to context everywhere and $\text{coeffs}$ to refer to the coefficients. The only place we can find where we use $c$ to refer to coefficients is in Appendix E where we use $a, b, c, d$ to refer to the cubic coefficients where the meaning should be clear from the usage
> - Use of the word causal: We agree this can be confusing and shall change causal/causality to physical/physicality, implying a physically plausible solution requires the future to not affect the past

---

> ### Author Response · Authors · 2023-06-30
> **On Incremental and Limited Innovations**
>
> It is unfortunate you believe the technical innovations are limited. As mentioned these were not supposed to be the main message of the paper, and have clearly diluted the main focus, which was to show that the right continuous model can outperform the current best discrete-time model on Multiple Sclerosis evolution prediction. We will significantly rewrite our introduction and contributions sections to make this clear. As we have said, using the interpolations to achieve parallel training in batches was a natural consequence of already having the interpolations to use for the Neural CDE model, and was not supposed to be viewed as our main contribution.
>
> That being said, we’d like to respectively point out that novelty and significance are not part of the TMLR acceptance criteria:
>
> 1. “TMLR emphasizes technical correctness over subjective significance, to ensure that we facilitate scientific discourse on topics that may not yet be accepted in mainstream venues but may be important in the future.” https://www.jmlr.org/tmlr/index.html
> 2. “Crucially, it should not be used as a reason to reject work that isn't considered 'significant' or ‘impactful’ because it isn't achieving a new state-of-the-art on some benchmark. Nor should it form the basis for rejecting work on a method considered not ‘novel enough’, as novelty of the studied method is not a necessary criteria for acceptance. We explicitly avoid these terms (‘significant’, ‘impactful’, ‘novel’), and focus instead on the notion of ‘interest’. If the authors make it clear that there is something to be learned by some researchers in their area from their work, then the criteria of interest is considered satisfied.” https://www.jmlr.org/tmlr/acceptance-criteria.html
>
> So provided our work is correct and of interest to the TMLR community, then novelty and significance should not be a major factor. We  believe we satisfy this criteria since the underlying methodology is correct - the results in Table 1 support this claim, and the results are still of interest to the TMLR audience. The practical application of Neural CDEs is still in its infancy, and this paper still remains a thorough investigation of how adjusting the methods leads to changes in experimental results. We strongly believe this is not only an interesting read, but also uncovers areas that need further exploration from the community. Specifically, we are showing that continuous time models are at least able to compete with discrete-time models when modeling Multiple Sclerosis evolution. But also that being continuous doesn’t automatically guarantee performance gains, good model selection and hyperparameter tuning must still be done to achieve the results. The results we obtained are consistent with Kidger et al. 2020 further supporting the use of Neural CDEs. And finally we tested two new interpolation schemes which did not outperform or underperform existing interpolation schemes; these results are consistent with Morrill et al. 2021, which shows the learnable part of the Neural CDE makes up for the interpolation scheme used.
>
> We would also like to point to all four reviewers agreeing and checking “Yes” to the Audience criteria, to support the suitability to TMLR despite the methodological contributions being limited compared to a methodology paper.
>
> # References
>
> 1. Kidger, P., Morrill, J., Foster, J. and Lyons, T., 2020. Neural controlled differential equations for irregular time series. Advances in Neural Information Processing Systems, 33, pp.6696-6707.
> 2. Morrill, J., Kidger, P., Yang, L. and Lyons, T., 2021. Neural controlled differential equations for online prediction tasks. arXiv preprint arXiv:2106.11028.

---

### Review · Reviewer_v21S · 2023-06-19

**Summary Of Contributions:**

The paper benchmarks the multiple sclerosis prediction tasks with several continuous time models and extensively experiments different interpolation methods. The paper shows that the continuous time model outperforms discrete time models, and standardizing features outperforms interpolating features.

**Audience:**

Yes

**Broader Impact Concerns:**

The paper proposes a model for healthcare prediction. The performance of the model should be carefully reviewed before being deployed in practice.

**Claims And Evidence:**

Yes

**Requested Changes:**

The authors should provide more details on the definition of the label and inclusion criterion for each tasks. Also, more insights on the models will be helpful, since the paper does not have contribution on methodology.

**Strengths And Weaknesses:**

Strength:

- There are sufficient ablation studies which demonstrates the contribution of each technical details on the performance.

- The literature review and comparison among different methods are clear and straight forward.

Weakness:

- The definition of label is not quite clear. Do the positive patients in 6-12 month also include patients who has already been positive in 0-6 month, or they are purely new on-set patients? The progression may be better defined in the framework of survival analysis instead of classification.

- Lack of novelty. The paper only practices several existing methods on a new task. Also, the analysis on what leads to such discrepancy among different models is insufficient.

---

> ### Author Response · Authors · 2023-06-30
> **Official Response to Reviewer v21S**
>
> We would like to start by thanking the reviewer for their time and effort to review our paper. We are glad that the reviewer found the literature review clear & straightforward and the ablation studies sufficient. We will happily answer your concerns summarised below:
>
> 1. Label definition
> 2. Novelty and Model Comparisons
> 3. Possible Broader Impact Concerns
>
> We will reply to these in separate responses so discussion is distinct. We will also update the manuscript and inform all reviewers when it is uploaded.

---

> ### Author Response · Authors · 2023-06-30
> **1. Label Definition**
>
> We have a collection of subjects, each have EDSS scores that are provided by clinicians at every visit. So if we consider a single subject we have a time-series of their EDSS scores. Following this, our modeling aim is to predict what the EDSS score will be a particular distance into the future. So for each $t_i$ in the time-series the modeling task is to predict the EDSS score at $t_i + \text{6 months}$, if 6 months is an example prediction horizon. Due to the irregularity of visits these prediction horizons are split into windows to account for this, 0-6 months, 6-12 months, 12-18 months, 18-24 months, and over that window any EDSS scores taken in clinic are averaged to give the label for the prediction horizon at that point in the time-series.
>
> The tasks are predicting the EDSS score directly, and also some clinically insightful tasks, EDSS > 3, EDSS > 5, and predicting the EDSS category. We do not use new patients for the different windows or tasks, so yes a patient with a certain score in the 0-6 month window is included in the 6-12 month window. As we describe, the models can only use the past to predict the future, and so it would not have access to the 0-6 month information when predicting about the 6-12 month window. We do not believe survival analysis is as appropriate as our setting here, firstly so that we can directly compare to Roy et al. 2022, and secondly because the EDSS score is not considered a discrete event, which ends the time-series which is more relevant to survival analysis, rather a dynamical value that changes over time and we are predicting the future value. A very good visual representation is Figure 3 in Roy et al. 2022 to show how the label is determined.
>
> We shall add this description to the manuscript.
>
> # References
>
> 1. Roy, S., Mincu, D., Proleev, L., Rostamzadeh, N., Ghate, C., Harris, N., Chen, C., Schrouff, J., Tomašev, N., Hartsell, F.L. and Heller, K., 2022, April. Disability prediction in multiple sclerosis using performance outcome measures and demographic data. In Conference on Health, Inference, and Learning (pp. 375-396). PMLR.

---

> ### Author Response · Authors · 2023-06-30
> **2. Novelty and Model Comparisons**
>
> Firstly we’d like to respectively point out that novelty is not part of the TMLR acceptance criteria:
>
> 1. “TMLR emphasizes technical correctness over subjective significance, to ensure that we facilitate scientific discourse on topics that may not yet be accepted in mainstream venues but may be important in the future.” https://www.jmlr.org/tmlr/index.html
> 2. “Nor should it form the basis for rejecting work on a method considered not ‘novel enough’, as novelty of the studied method is not a necessary criteria for acceptance. We explicitly avoid these terms (‘significant’, ‘impactful’, ‘novel’), and focus instead on the notion of ‘interest’. If the authors make it clear that there is something to be learned by some researchers in their area from their work, then the criteria of interest is considered satisfied.” https://www.jmlr.org/tmlr/acceptance-criteria.html
>
> So provided our work is correct and of interest to the TMLR community then novelty should not be a major factor. Since the practical application of Neural CDEs is still in its infancy, and this paper still remains a thorough investigation of how adjusting the methods leads to changes in experimental results. We strongly believe this is not only an interesting read, but also uncovers areas that need further exploration from the community.
>
> Regarding the differences in models, we do address this in the paper, “Neural CDE outperforming ODE-RNN, Latent ODE and GRU-ODE is consistent with the findings in (Kidger et al., 2020), we hypothesize this is due to Neural CDEs having provably more representation power acting on sequences than the other continuous models.” Our findings for model discrepancy are supported by Kidger et al. 2020, which contains the theorem saying any ODE given by $\frac{dz}{dt}= f_\theta(z, X_t)$ can be represented exactly as $\frac{dz}{dt} = \tilde{f}_\phi(z)\frac{dX_t}{dt}$ however the converse is not true. Kidger et al. 2020 make the same claim “In our experiments, we find that the Neural CDE substantially outperforms the GRU-ODE, which we speculate is a consequence of this result.”
>
> Following this we carry out some ablations looking at which interpolation methods work best. We find that the choice of interpolation is not a significant factor. This is consistent with Morrill et al. 2021 which was the first paper to introduce multiple spline choices for Neural CDEs. They found that the choice of interpolation method did not significantly affect results, suggesting the learnable part of the Neural CDE dynamics accounts for the different interpolations.
>
> The final ablation in the main body of our paper investigates standardization vs interpolation. We found that standardization is more effective for Neural CDEs. We hypothesize that this is because standardization brings $dX/ds$ into smaller ranges, making the dynamics more stable. This is supported by the same ablations performed on the other continuous models in Appendix B.1. We write, “for these three methods the differences are small, whereas with Neural CDEs there was a clear distinction between standardizing and not standardizing features. This is likely due to the models’ dynamics functions. The dynamics function for the Neural CDE ends with $dX/ds$, whereas all others end with $d\psi/ds$. The true times are relatively small and therefore so is $d\psi/ds$. However, $X$ can be quite large, and therefore so is $dX/ds$. And so standardizing $X$ aids Neural CDEs more because $dX/ds$ in the dynamics is explicitly shifted to a more manageable range, whereas $d\psi/ds$ was already in that range”.
>
> We finally point out that whilst not the main focus, the work does have some small technical contributions. We introduce two splines that can be used as control signals that complement those introduced in Morrill et al. 2021, the monocubic and recticubic splines. We took the method of batching irregular time-series in Morrill et al 2021. (using the tree-invariance property of Neural CDEs) and generalized for simpler implementation applied to all ODE models, by letting $t = \psi(s)$ where $s$ is a regular set. Finally we adapted the framework from Roy et al. 2022 to include new preprocessing steps in the pipeline to allow continuous models to be used when modeling MS.
>
> # References
> 1. Kidger, P., Morrill, J., Foster, J. and Lyons, T., 2020. Neural controlled differential equations for irregular time series. Advances in Neural Information Processing Systems, 33, pp.6696-6707.
> 2. Morrill, J., Kidger, P., Yang, L. and Lyons, T., 2021. Neural controlled differential equations for online prediction tasks. arXiv preprint arXiv:2106.11028.
> 3. Roy, S., Mincu, D., Proleev, L., Rostamzadeh, N., Ghate, C., Harris, N., Chen, C., Schrouff, J., Tomašev, N., Hartsell, F.L. and Heller, K., 2022, April. Disability prediction in multiple sclerosis using performance outcome measures and demographic data. In Conference on Health, Inference, and Learning (pp. 375-396). PMLR.

---

> ### Author Response · Authors · 2023-06-30
> **3. Possible Broader Impact Concerns**
>
> We fully agree with that given the suggested application a model needs to be tested to a greater extent before being deployed. We believe it should be scrutinized to at least the same standard as is used in drug trials. We do not claim that the model is ready to be deployed and in our Broader Impact Statement after the conclusion we have discussed this, “Since this work is in early stages it should not be deployed yet, and more research and
> evaluation is required first. If deployed, it must be used alongside the expert opinion of trained practitioners, and if in disagreement a further expert assessment is sought out.“ We shall edit this to say that the model *must* not be deployed until it undergoes further testing. And if there is disagreement between the model and the trained experts a further expert assessment should be sought out, and that the expert's opinion currently is more reliable than the model’s predictions.

---

### Review · Reviewer_QEEx · 2023-06-29

**Summary Of Contributions:**

The paper presents an approach to predicting the progression of multiple sclerosis (MS) modeling changes through continuous time via the NODE. The framework adapts existing continuous models to handle irregular and missing data through a preprocessing step and two new interpolation schemes. The preprocessing step standardizes sequences, imputes missing context features, pads sequences to the same length, and interpolates the new causal time-series using integers as pseudo-time. The interpolation schemes, including a monotonic cubic scheme for the time channel, are designed to interpolate missing values and act as continuous control signals for models. The authors demonstrate that this approach, which intrinsically handles irregular time-series and imputes missing values with physically plausible values (smooth), often outperforms a discrete model in benchmark tests on a public MS dataset. The paper underscores the potential of continuous models and machine learning methods in predicting MS progression and informing interventions and treatment options for patients.


**Audience:**

Yes

**Broader Impact Concerns:**

No concerns

**Claims And Evidence:**

No

**Requested Changes:**

1.  Justify why is this an ML paper rather than an application paper suitable to a specific venue (this is the most important concern)
2.  Perform a thorough comparison with proper classical and modern time series forecasting models in the same settings with and without the interpolation schemes.
3.  Better explain the differences with the CHIL paper in the text
4.  Address weakness #4 above.


**Strengths And Weaknesses:**

# Strengths

1.  _The problem addressed by the work is of high importance_
2.  The improvement over the results shown in the paper published in CHIL 2022 are notable (however see weakness #1 )
3.  A detailed exposition of the techniques used in the experiments. I would conclude that the paper is well structured if not for weakness #4.


# Weaknesses

1.  *Inconclusive demonstration of performance of the proposed approach.* It is unjustifiably assumed that TCN is the state of the art model and comparison with it demonstrates the point. However, checking the previous paper that this work extends, one can see that the competing models are all weak (neither is specifically a time series model). Thus, beating TCN does not seem to lead to any conclusions about performance of the NeuralCDE model investigated in this paper. *The claims of outperforming the current state of the art are unjustified.* How about looking into classical forecasting time series models and their ensembles (see a couple of links below for the starting point)? Maybe look into Cyclic Boosting as well? I am not sure why this is not done.
    -   [Example classical forecasting models](https://github.com/Nixtla/statsforecast/tree/main/experiments/m3)
    -   [a package for forecasting models](https://github.com/Nixtla/statsforecast/tree/main)
2.  *Highly specific to the proposed application.* Although the application to MS problem is extremely important, the paper, despite being technical, feels more appropriate in a specialized venue than at TMLR.
3.  *Limited innovation.* Most of the ideas used in the paper, including the interpolation to create continuous signal, have already been published in the preceding paper. The main novelty is in application of NODE approach to the POM data for MS prediction, while previously it was a TCN. The elaborate interpolation schemes did not show a large effect in the ablation study, leaving the performance gain to the use of NODE.
4.  Sufficiently convoluted writing. This is minor, but definitely makes the paper harder to read. Some of the terms appear before their definitions.
    1.  Mathematical notation appears in the text without sufficient explanation of the meaning, e.g. "c" on page 4, "recti" on page 4 in item 2, what are the weights w, etc.?
    2.  The experiments are not sufficiently described. How is the forecasting problem turned into classification? I guess that a prediction produced by the model at each time point is compared with the actual value or EDSS and each such comparison is considered as a classification event, but I am unable to confirm this in the paper besides asking the authors. Do the authors have EDSS available even where measurements are missing? Many unanswered questions

---

> ### Author Response · Authors · 2023-06-30
> **Official Response to Reviewer QEEx**
>
> Firstly we would like to express our sincere thanks for taking the time and effort to review our paper. We are glad you found the problem important, the results improve on the Roy et al. 2022 baseline and that the paper was mostly well written. We are grateful for the detailed feedback and requested changes summarized as:
>
> 1. Application heavy paper
> 2. Further comparisons
> 3. Differences with Roy et al. 2022
> 4. Clarity of writing
>
> We answer all these points below in separate replies to keep discussion distinct. We shall also update the manuscript and inform all reviewers when this is uploaded.

---

> ### Author Response · Authors · 2023-06-30
> **1. Application Heavy Paper**
>
> Indeed our paper’s main contribution is to look at the use of continuous models for modeling Multiple Sclerosis progression. Since the review period began, we have realised our introduction is confusing and dilutes the main message of the paper by also framing other contributions as more significant than we would have liked. We will edit the introduction to reduce these claims, so that our primary contribution is benchmarking four continuous models showing they can compete with discrete-time models, and our secondary contributions are the technical ones that made it possible (new preprocessing pipeline, batching method, interpolations). So why is this paper suitable for TMLR? To answer this we respectively quote the TMLR acceptance criteria https://jmlr.org/tmlr/acceptance-criteria.html:
>
> “Acceptance of a submission to TMLR should be based on positive answers to the following two questions:
>
> 1. Are the claims made in the submission supported by accurate, convincing and clear evidence?
> 2. Would some individuals in TMLR's audience be interested in the findings of this paper?”
>
> And in particular:
>
> “Crucially, it should not be used as a reason to reject work that isn't considered 'significant' or 'impactful' because it isn't achieving a new state-of-the-art on some benchmark. Nor should it form the basis for rejecting work on a method considered not 'novel enough', as novelty of the studied method is not a necessary criteria for acceptance. We explicitly avoid these terms ('significant', 'impactful', 'novel'), and focus instead on the notion of 'interest'. If the authors make it clear that there is something to be learned by some researchers in their area from their work, then the criteria of interest is considered satisfied. TMLR instead relies on certifications (such as 'Featured' and 'Outstanding') to provide annotations on submissions that pertain to (more speculative) assertions on significance or potential for impact.”
>
> The suitability criteria is covered by the second point, which is: if some of the TMLR audience would find the work interesting, or if there is something to be learned it meets the criteria, it does not exclude application papers. The practical application of Neural CDEs is still in its infancy, and this paper still remains a thorough investigation of how adjusting the methods leads to changes in experimental results. We strongly believe this is not only an interesting read, but also uncovers areas that need further exploration from the community. Specifically, we are showing that continuous time models are at least able to compete with discrete-time models when modeling Multiple Sclerosis evolution. But also that being continuous doesn’t automatically guarantee performance gains, good model selection and hyperparameter tuning must still be done to achieve the results. The results we obtained are consistent with Kidger et al. 2020 further supporting the use of Neural CDEs. And finally we tested two new interpolation schemes which did not outperform or underperform existing interpolation schemes; these results are consistent with Morrill et al. 2021, which shows the learnable part of the Neural CDE makes up for the interpolation scheme used.
>
> We would also like to point to all four reviewers agreeing and checking “Yes” to the Audience criteria, to support the suitability to TMLR despite the methodological contributions being limited compared to a methodology paper.
>
> # References
>
> 1. Kidger, P., Morrill, J., Foster, J. and Lyons, T., 2020. Neural controlled differential equations for irregular time series. Advances in Neural Information Processing Systems, 33, pp.6696-6707.
> 2. Morrill, J., Kidger, P., Yang, L. and Lyons, T., 2021. Neural controlled differential equations for online prediction tasks. arXiv preprint arXiv:2106.11028.

---

> ### Author Response · Authors · 2023-06-30
> **2. Further Comparisons**
>
> Firstly, we agree that framing TCN as the state of the art is misleading. We have changed all points in the manuscript saying this to instead describe TCN as the best performing model from Roy et al. 2022, or the previously best benchmarked discrete model. However, we also believe that the benchmarked models in Roy et al. 2022 cover a wide range of possible models. There is a model from each major category of techniques: logistic and linear models representing classical models; gradient boosted classifiers/regressors representing tree based methods; MLP as a simple deep learning approach and TCN as a deep learning approach for sequences.
>
> We did test RNNs initially however we found they were noticeably worse than TCN so did not include them in our results. Would it be sufficient to reduce our claims to say that we benchmark four continuous models, and that Neural CDEs specifically are able to beat the best discrete model tested in this setting? We believe this would satisfy the TMLR criteria, since the underlying methodology would be correct - the results in Table 1 support this claim; and the results would still be of interest to the TMLR audience. We think it would be more realistic and honest to reduce our claims rather than promise to benchmark further models and not deliver rigorous results in the short two week reviewing period.
>
> # References
>
> 1. Roy, S., Mincu, D., Proleev, L., Rostamzadeh, N., Ghate, C., Harris, N., Chen, C., Schrouff, J., Tomašev, N., Hartsell, F.L. and Heller, K., 2022, April. Disability prediction in multiple sclerosis using performance outcome measures and demographic data. In Conference on Health, Inference, and Learning (pp. 375-396). PMLR.

---

> ### Author Response · Authors · 2023-06-30
> **3. Differences with Roy et al. 2022**
>
> Roy et al. 2022 laid the foundations to carry out this work. They took the Multiple Sclerosis datasets and placed them all into a common representation. Created a preprocessing pipeline to convert the common representation into a format that can be used by various machine learning models, the preprocessing is different for different models. They benchmarked a collection of models, ranging from linear models to TCN. Finally they calculate the metrics separately for subgroups of the subjects to determine which groups can be more susceptible.
>
> Our work adds to the preprocessing pipeline, by adding a part that converts the data to a format that can be used by continuous time models, this is given in Section 3.1. We also added to the modeling part, where we have benchmarked four continuous time models (described in Section 4), under the motivation that continuous time may be a more realistic inductive bias to enforce, as well as naturally handling irregular time-series. We benchmarked the models and saw that Neural CDE tends to outperform TCN whereas the other three models do not. In Roy et al. 2022 Figure 1 shows the whole pipeline, showing their contributions, we added to the third (green) and fourth (purple) parts of this figure.
>
> As well as this, we tested the idea of interpolating the time-series to calculate missing features, this was not done in Roy et al. 2022, the idea originates from Kidger et al. 2020. We found that this did not lead to a significant difference, but it was still worthwhile testing, and is a valuable result to the community. We introduced two new interpolation schemes for Neural CDE, again after testing we did not see them outperform or underperform any other interpolation schemes but it was still a worthwhile test so that now it is known, it also supports results from Morrill et al. 2021. We finally generalized the idea from Morrill et al. 2021 of using the interpolation to train in batches to other continuous models. This was used since the interpolations were already calculated for the Neural CDE and was a natural solution to train in batches when we did not have a parallel differentiable solver available to us (we were using the TensorFlow solver https://www.tensorflow.org/probability/api_docs/python/tfp/math/ode/Solver).
>
> Overall we would argue that this paper differs significantly from both Roy et al. 2022 and Kidger et al. 2020. While it combines techniques from both, and builds upon results from both, it is the intersection of the theoretical nature of continuous time modeling and the practical application of healthcare where we believe our work shines.
>
> # References
>
> 1. Roy, S., Mincu, D., Proleev, L., Rostamzadeh, N., Ghate, C., Harris, N., Chen, C., Schrouff, J., Tomašev, N., Hartsell, F.L. and Heller, K., 2022, April. Disability prediction in multiple sclerosis using performance outcome measures and demographic data. In Conference on Health, Inference, and Learning (pp. 375-396). PMLR.
> 2. Kidger, P., Morrill, J., Foster, J. and Lyons, T., 2020. Neural controlled differential equations for irregular time series. Advances in Neural Information Processing Systems, 33, pp.6696-6707.
> 3. Morrill, J., Kidger, P., Yang, L. and Lyons, T., 2021. Neural controlled differential equations for online prediction tasks. arXiv preprint arXiv:2106.11028.

---

> ### Author Response · Authors · 2023-06-30
> **4. Clarity of Writing**
>
> We apologize that part of the writing is not as clear as we would have liked. We do already introduce $\mathbf{c}$ at the start of page 4, “At each timestamp we have a vector of sequence features $\mathbf{x}$, a vector of context features $\mathbf{c}$, a timestamp $t$, a label $y$ and a sample weight $w$. The sample weights inform the loss function and evaluation metric if a prediction is used in the calculation.” To be more clear, $\mathbf{c}$ are context features, described on page 3, “the context gives static information that does not change over time, such as sex and ethnicity”. And $w$ are sample weights, which tell the model a label’s contribution to the loss calculation. That is, $w$ is a time-series consisting of 0s and 1s, the same length as the time-series, if $w_i$ is 1 then $y_i$ and the predicted $\hat{y}_i$ are used in the loss calculation. This allows us to pad the sequences to the same length to train easily in batches. It also allows us to handle missing labels, if we lack a label we impute with 0 and set $w$ to 0 as well. We describe the recti method relatively soon after mentioning it which is the “Causal Interpolation” paragraph. We have firstly renamed this to be the “Causal Interpolation (Recti method)” paragraph, and before its introduction we shall name it causal interpolation rather than recti. We shall rewrite parts of page 4 to make this more readable.
>
> Regarding the experiments and specifically the labels, we have a collection of subjects, each have EDSS scores that are provided by clinicians at every visit. So if we consider a single subject we have a time-series of their EDSS scores. Following this, our modeling aim is to predict what the EDSS score will be a particular distance into the future. So for each $t_i$ in the time-series, the modeling task is to predict the EDSS score at $t_i + \text{6 months}$, if 6 months is an example prediction horizon. Due to the irregularity of visits these prediction horizons are split into windows to account for this, 0-6 months, 6-12 months, 12-18 months, 18-24 months, and over that window any EDSS scores taken in clinic are averaged to give the label for the prediction horizon at that point in the time-series. If there are still missing labels for this horizon we impute with 0 and set $w$ to 0 so it is not used in the loss calculation.
>
> The tasks are predicting the EDSS score directly, and also some clinically insightful tasks. These are: predict if EDSS > 3, predict if EDSS > 5 and predict the EDSS category. As mentioned in the dataset description at the start of Section 5, EDSS is a score from 0 - 10 in increments of 0.5. EDSS scores are partitioned into four distinct categories: 0 - 1 for no disability, 1.5 - 2.5 for mild disability, 3 - 4.5 for moderate disability, and 5 - 10 for severe disability. This is how we can convert the task into a classification task, and due to the existence of these categories, the classes in the classification problems are clinically insightful hence why we use them (as well as being able to compare to Roy et al. 2022). We shall add the label description to the main paper.
>
> # References
>
> 1. Roy, S., Mincu, D., Proleev, L., Rostamzadeh, N., Ghate, C., Harris, N., Chen, C., Schrouff, J., Tomašev, N., Hartsell, F.L. and Heller, K., 2022, April. Disability prediction in multiple sclerosis using performance outcome measures and demographic data. In Conference on Health, Inference, and Learning (pp. 375-396). PMLR.

---

### Review · Reviewer_7QTB · 2023-06-29

**Summary Of Contributions:**

Contribution proposes a kind of reparametrization in the context of NeuralODE to have efficient batching during learning with irregular time series
By comparing different NeuralODE solvers on the task of MS disease progression prediction, authors show that Neural Controlled Differential Equations can outperform causal discrete models like temporal convolutional networks (TCN). The 3 other NeuralODE solvers do not perform as well.
Also to deal with continuously defined data given discrete inputs in a prediction context, smooth and causal interpolations are investigated. The recticubic method is promoted.
The work builds heavily on Roy et al. (2022)

--
Sorry for the late review, I had not submitted it in the right place...

**Audience:**

Yes

**Claims And Evidence:**

Yes

**Requested Changes:**

The paper requires a good practical experience with NeuralODE solvers to be fully grasped. Certain technical paragraph should expanded and rewritten with a bit more pedagogy:

“And then at observations undergoes a discrete change.” Can you rephrase?
“This evolves continuously between observations and at observations changes discretely based on its current value, the observation and context using gφc .” this needs rephrasing too. The full paragraph would benefit from editing to be honest.
The paper claims some computational gains but no details or comparison in running time is presented. I would suggest to be more explicit on computational costs.

Also I would suggest to add a pseudo-code of the best method to be able to follow what are input parameters, how data are batched, how gradient updates are done etc.

I would appreciate some code in supplementary material to be able to experiment myself with the approach.

Typos:

“biological mechanisms so “ -> biological mechanisms “
It is critical that a model is casual -> It is critical that a model is causal
For example,this -> For example, this
Note the Latent ODE … -> Note that the Latent ODE
an further expert -> an expert

**Strengths And Weaknesses:**

See cell below

---

> ### Author Response · Authors · 2023-06-30
> **Official Response to Reviewer 7QTB**
>
> Many thanks for the time and effort in reviewing our paper. We are grateful for the constructive feedback summarized below:
>
> 1. Clearer Writing
> 2. Claimed Computational Gains
> 3. Pseudo-Code
> 4. Code
> 5. Small typos
>
> We address all of the points in separate replies to keep discussion distinct. We will also update the manuscript and inform all reviewers when this has been uploaded.

---

> ### Author Response · Authors · 2023-06-30
> **1. Clearer Writing**
>
> We are happy to rewrite any parts of the manuscript to make them clearer. We shall add a part about Neural ODEs and ODE solvers. In particular, that one way to solve ODEs is with the Euler solver:
>
> $z_{t + \Delta t} = z_t + f(z_t, t)\Delta t$
>
> The output is given by first inputing the initial condition $z(t_0)$ and repeatedly applying this update rule. The output of this solver approaches the true ODE solution as $\Delta t$ approaches 0. Additionally, this is the most simple ODE solver and that there are many more advanced solvers from the numerics community, such as Runge-Kutta and adaptive solvers such as the Dormand-Prince solver which we use.
>
> Regarding the paragraph describing the ODE-RNN, all paragraphs in Section 4 are generally about the models, and are best read alongside Figure 1, which shows the models. Additionally the mathematical formulation for each model is given directly below its description in order of the steps taken. We have changed “discrete change” to “discontinuous change” everywhere, since it is more representative of what happens. That is, when we reach an observation, the hidden state undergoes a finite change in zero time, and so is a discontinuous change. In the same way at observations RNNs’ hidden states undergo discontinuous changes. As well as this we have added a sentence pointing to Figure 1, “See the top right of Figure 1 to see how the state changes continuously between observations and discontinuously at observations”. We have also added a sentence pointing to the mathematical formulation beneath, “see the equations of each model below their description”.

---

> ### Author Response · Authors · 2023-06-30
> **2. Claimed Computational Gains**
>
> We do not believe that experimental comparisons are necessary in this case, since the speed and memory requirements are supported both by existing literature and the asymptotic complexities of the methods.
>
> On page 60 of Kidger 2022, Section 3.2.1.3, Kidger remarks while discussing a related scheme for Neural CDEs: “Remark 3.12. This was actually a mistake we made in [Kid+20a] – the above procedure was not done, in favour of an alternate (storage-inefficient) scheme that involved taking the union over the times $t_j$ needed to handle each batch of data.” That is, instead of the method that we propose (which is a generalisation from Morrill et al 2021.), the times for each series in the batch are merged & sorted, then the combined system is solved for all times and only certain outputs are used in the loss calculation using a binary mask. So the existing literature supports this method over the union method of batching.
>
> We can also show this by considering the complexities of the two methods. In the following, assume that all time-series in the dataset are length $n_t$, the analysis can be straightforwardly adapted when this is not the case. Assume that all initial times are the same, which is a requirement of merging the time-series to train in batches (our method does not require this). We also assume that no two time-series share any evaluation times, if they do then these are combined when merging the sorted lists, the complexity is still the same.
>
> We start by considering the method of taking unions. First the union must be taken over sets of times in the batch and sorted. To merge the sorted lists is $\mathcal{O}(n_b n_t)$ where $n_b$ is the batchsize. As well as this there is a memory cost of $\mathcal{O}(n_b^2 n_t)$ which is a binary mask to indicate whether a result at a given time for a given series in the batch is relevant for the loss calculation, i.e. for each series in the batch we need a binary mask of length $n_b n_t$, the total number of merged times. During the solve we calculate the output of the ODE at $n_b n_t$ times, which means during the backward solve, we have to stop the solve explicitly this many times to adjust the adjoint state (which undergoes discontinuous jumps at measurement points, see Chen et al. 2018 for an explanation of this). This slows down the backward solve significantly since there is a cost of restarting an ODE solve. Assuming that during training there is shuffling of the series to create new batches each epoch, then this must be done for every iteration of training.
>
> In contrast, the method of solving the ODE in a related domain leads to the following speed ups. To calculate the interpolation $\psi$, we have to interpolate each time-series individually which gives cost $\mathcal{O}(n_s n_t)$ where $n_s$ is the total number of subjects. Note $n_s$ is the sum of all the batch sizes from a single epoch, however this is a single preprocessing step so does not need to be done at each training iteration, therefore assuming we train for more than one epoch, interpolating is faster. Regarding the memory complexity, we need to now store cubic coefficients of the interpolation. This gives four coefficients for each point in each series giving memory complexity $\mathcal{O}(4n_s n_t)$. The memory complexity of storing the binary mask in the alternative method was $\mathcal{O}(n_b^2 n_t)$ which, depending on batch size, will often be more memory inefficient (for a train set of size 10,000 a batch size of 200 is less efficient), as noted in Kidger 2022 above, this is often the case. Finally during the solve we only need to solve for $n_t$ times, which during the backwards solve of the adjoint method provides significant speedup. We will add this analysis and the citation of Kidger 2022 to the explanation of our choice to batch in this way.
>
> We also accept that our contributions section has made the paper confusing. Our main contribution is testing continuous time models on a real MS dataset to show that if a continuous model is well tuned it can outperform discrete time models, paving the way for further research. The method of using $\psi$ to parallelize the batches was used as a high level solution to train in batches as a natural second use of the existing interpolation (already calculated for the Neural CDE). The new interpolation and batching methods are contributions but we do not want that to dilute the main message so shall rewrite our introduction to make this point.
>
> # References
>
> 1. Kidger, P., 2022. On neural differential equations. arXiv preprint arXiv:2202.02435.
> 2. Morrill, J., Kidger, P., Yang, L. and Lyons, T., 2021. Neural controlled differential equations for online prediction tasks. arXiv preprint arXiv:2106.11028.
> 3. Chen, R.T., Rubanova, Y., Bettencourt, J. and Duvenaud, D.K., 2018. Neural ordinary differential equations. Advances in neural information processing systems, 31.

---

> ### Author Response · Authors · 2023-06-30
> **3. Pseudo-Code**
>
> We’ll happily add pseudo-code for the Neural CDE. Our hope was that the whole paper, in particular Sections 3 and 4, would act as a surrogate extended pseudo-code to help the reader follow the procedure. The procedure is as follows:
>
> - We receive a dataset, and split the subjects into train and test subjects, to avoid data leakage
> - We individually preprocess the train and test datasets in accordance with Section 3.1, this creates a padded dataset where for each subject the time-series is of the form $((\text{coeffs}, \mathbf{c}), y, w)$. Where coeffs are a series of cubic coefficients that give a piecewise interpolation of the features. $\mathbf{c}$ is a vector of context features that do not change such as sex and ethnicity. $y$ is the label in the form of a time-series. $w$ is a time-series giving sample weights, which inform the model whether to track the prediction at this time in the loss, i.e. it is a vector of 0s and 1s the same length as $y$. This is constructed when we do padding.
>
> During training, for each epoch:
> - Shuffle the dataset along the batch dimension, and construct minibatches from this shuffled train set
> - Encode the initial condition and context $h_0 = e(X(0), \mathbf{c})$
> - Use the coeffs to calculate the interpolation $X(s)$ or the derivative $dX/ds$
> - Solve the hidden NCDE $\frac{dh}{ds} = f(h, c)\frac{dX}{ds}$, $h(0) = h_0$
> - Predict the label from the hidden latent state $\hat{y}_i = d(h_i)$
> - Calculate the loss using $y_i$, $\hat{y}_i$ and $w_i$
> - Backpropagate using the adjoint method and update the parameters with a given optimizer

---

> ### Author Response · Authors · 2023-06-30
> **4. Code**
>
> Currently our code is not publicly available, we do plan to open source in the future. This is because our code does not exist in isolation but is part of a larger codebase containing unpublished/unsubmitted work. We include full implementation details in the Appendix for reproducibility purposes. We do include some code in Appendix F.2 to demonstrate how to include repeated calls to an ODE solver. If there is a specific section of code (for example the implementation of a model) that you would like to request we might be able to include that.

---

> ### Author Response · Authors · 2023-06-30
> **5. Typos**
>
> Many thanks for pointing these out, we have now fixed them.
>
> The one exception is:
>
> “In our case the progression of MS is driven by biological mechanisms so can be modeled with differential equations.”
>
> Has been changed to:
>
> “In our case the progression of MS is driven by underlying biological mechanisms so MS progression can also be modeled with differential equations.”
>
> That is, since Multiple Sclerosis is driven by underlying biological and chemical mechanisms, we can model the evolution of Multiple Sclerosis using differential equations to define the time evolution of a latent state from which the observations are realised.

---

### Author Response · Authors · 2023-07-07
**Updated Manuscript Part 1/2**

To all reviewers and action editors,

In line with your constructive feedback and requested changes, we have updated and uploaded our manuscript including many of the requested changes. The changes are summarized below:

# Major Changes
- Changed mention of TCN being state of the art everywhere to the current best benchmarked discrete time model. We appreciate framing TCN as SOTA is not representative and have changed that (QEEx)
- Added a reference in the main paper Section 3 pointing to Appendix C, where we now further explain the computational and memory complexities of our proposed batching method against the method of taking unions. We include the citation of Kidger 2022, showing the use of pseudo-time over unions is supported by existing literature, as well as the asymptotic complexities showing why pseudo-time is better in theory. We also explicitly say that this was not experimentally tested (7QTB, eSNb)
- Improved the explanation of labels in the dataset at the start of Section 5, to include the discussion from our responses, this also includes why and how the task can be formulated as a classification (v21S, QEEx)
- Made major changes to reframe the paper and reduce the claims. We have reflected on all reviewers comments, and understand that the claims need to be reduced. As such, we have changed our title to make it clear the main contribution is benchmarking four continuous models on this dataset. We have also edited the abstract and rewritten the introduction & conclusion to frame our primary contributions as benchmarking the models and testing the use of interpolations; and that our secondary contributions are two new interpolation schemes and batching with pseudo-time (which was a technical solution to a common hurdle that did not require a low-level solver being implemented and repurposes already calculated interpolations from the Neural CDE) (7QTB, v21S, eSNb)

# Minor Changes
- Added description of the Euler Solver to Section 2 to provide more information to the reader unfamiliar with differential equations (7QTB)
- Edited the ODE-RNN paragraph to make continued reference to Figure 1 and the equations. As well as moved the update mask to the next paragraph, describing it as a technical detail. This has improved the clarity of the ODE-RNN explanation (7QTB)
- Edited the broader impact to say the method must not be deployed yet, it needs to undergo more testing similar to the scrutiny of a drug trial and that trained medical professionals are currently more reliable (v21S)
- Added to the final paragraph of Section 3.1 to describe how we can now train in minibatches with an optimizer of choice as is standard in Deep Learning. We believe this is effective in place of pseudo code of the training procedure (7QTB)
- Fixed typos (7QTB)
- Added further information to Section 3.1 when introducing sample weights (QEEx)
- Added further clarity around the term “recti” not using it until it is properly introduced, as well as referencing the sections in the paper more clearly where information can be found (QEEx)
- Introduced context features at the start of Section 3 more clearly (QEEx)
- Added the requirements of $\psi$ to Section 3 and Appendix C: increasing, surjective and piece-wise differentiable (eSNb)
- Added a sentence to the end of Section 5 (interpolation scheme ablation - Table 3) to make it clear we are testing the two methods of enforcing solutions that don’t use future measurements (eSNb)
- Changed the use of “causal” to describe models that only use past and current observations to make predictions to “physically plausible” (eSNb)
- Referenced Appendix B and B.3 more clearly to explain why standardizing features helps, as well as evidence of why this is the case and showing the existing feature ablations (eSNb)
- Added discussion of torchode (Lienen 2022) and diffrax (Kidger 2022), to Section 3 when discussing batching using pseudo-time. We mention how these are library specific, low level solutions, whereas our solution is library agnostic, high level and is reusing interpolations that have already been calculated for the Neural CDE (eSNb)
- Added to the conclusion and Section 5 the reasons Neural CDE is better than the other models (due to the expressive power of the model). And also that despite not all continuous models outperforming TCN, we have gained insight that we still need to choose the best model, and further supported the findings and usage of Neural CDEs (v21S, eSNb)
- Removed the suggestion of using the recticubic control signal since there is insufficient evidence to suggest it is the best (eSNb)

---

> ### Author Response · Authors · 2023-07-07
> **Updated Manuscript Part 2/2**
>
> Having implemented many of the suggested changes, we do believe that the paper has improved and in particular is more appropriate for acceptance at TMLR. With that, we point to the TMLR acceptance criteria (https://jmlr.org/tmlr/acceptance-criteria.html):
>
> 1. “Are the claims made in the submission supported by accurate, convincing and clear evidence?... Another way to satisfy this is simply for the authors to adjust (reduce) their claims.”
> 2. “Would some individuals in TMLR's audience be interested in the findings of this paper?... Crucially, it should **not** be used as a reason to reject work that isn't considered ‘significant’ or ‘impactful’ because it isn't achieving a new state-of-the-art on some benchmark. Nor should it form the basis for rejecting work on a method considered not 'novel enough', as novelty of the studied method is not a necessary criteria for acceptance. We explicitly avoid these terms (‘significant’, ‘impactful’, ‘novel’), and focus instead on the notion of ‘interest’. If the authors make it clear that there is something to be learned by some researchers in their area from their work, then the criteria of interest is considered satisfied.”
>
> The most significant changes we introduced were to rewrite parts of the manuscript and the title to reduce our claims. Now the main message of the paper is that we are benchmarking continuous models on this MS dataset and comparing to the previous best benchmarked model (TCN), we have removed any SOTA claims. We lower our technical contributions to *secondary* contributions, those are the continuous preprocessing pipeline, batching with pseudo-time and two new interpolation schemes. This is because they were technical hurdles requiring solutions to test the Continuous Models within the Roy et al. 2022 pipeline. In particular, we appreciate the claims about pseudo-time batching are not fully supported, and we clarify that this was a solution to the batching problem, that was high level (not requiring a low-level ODE solver implementation) that used already calculated interpolations from the Neural CDE. Additionally, we have further included citations and theory in Appendix C supporting this method over taking unions. Therefore, we believe that the claims are now supported by the evidence, we have benchmarked four continuous models with correct methodology, and our conclusion that Neural CDE can beat TCN is also supported.
>
> Despite our claims being reduced, we still strongly believe that the paper meets the second criteria. We have tested the use of continuous time models in a previously unexplored setting. We found that Neural CDEs achieve results that either beat or are competitive with the best benchmarked model (TCN), and we run extensive ablations to find the sources of gain. As mentioned, the practical application of Neural CDE models is still in its infancy, and this paper still remains a thorough investigation of how adjusting the methods leads to changes in experimental results. We strongly believe this is not only an interesting read, but also uncovers areas that need further exploration from the community.
>
> As such, we appreciate the time and effort of all reviewers already put in to make the paper better. We kindly request you review the changes made and consider our case for accepting the paper. As stated, we are already grateful for the time put in, the changes made have already improved the paper substantially, and if there are any further points that you wish to discuss we shall happily do so.
>
> # References
> 1. Kidger, P., 2022. On neural differential equations. arXiv preprint arXiv:2202.02435.
> 2. Lienen, M. and Günnemann, S., 2022. torchode: A Parallel ODE Solver for PyTorch. arXiv preprint arXiv:2210.12375.
> 3. Roy, S., Mincu, D., Proleev, L., Rostamzadeh, N., Ghate, C., Harris, N., Chen, C., Schrouff, J., Tomašev, N., Hartsell, F.L. and Heller, K., 2022, April. Disability prediction in multiple sclerosis using performance outcome measures and demographic data. In Conference on Health, Inference, and Learning (pp. 375-396). PMLR.

---

### Author Response · Authors · 2023-07-25
**Further Discussion?**

To the reviewers and the action editor,

The discussion period for this paper is scheduled to end in two days:

"You will have 2 weeks to respond to the reviewers. To maximise the period of interaction and discussion, please respond as soon as possible. The reviewers will be using this time period to hear from you and gather all the information they need. In about 2 weeks (Jul 13), and no later than 4 weeks (Jul 27), reviewers will submit their formal decision recommendation to the Action Editor in charge of your submission."

We have not received any replies to our responses, but we would be very happy to discuss any further points if needed. We really would like to interact and openly discuss the merits of our paper so please let us know if there are any remaining points.

To summarise, we have made the following *major* changes to the paper:

- We have reduced our claims so that the primary claims are benchmarking a range of continuous models on a publicly available Multiple Sclerosis dataset, and our secondary claims are the small technical novelties we used to make the implementation possible
- We have provided theoretical justification and existing citations to support the use of our technical secondary contributions, supporting both the space and time efficiency of the approaches
- We have changed our title and abstract to reflect these changes and be more faithful to the main contributions of the work

We believe with these changes we have satisfied the TMLR criteria (https://jmlr.org/tmlr/acceptance-criteria.html):

1. Are the claims made in the submission supported by accurate, convincing and clear evidence? (Any gap between claims and evidence should be addressed by the authors.)
2. Would some individuals in TMLR's audience be interested in the findings of this paper?

With our reduction in claims we believe we have now met the first criteria. We do not see any technical flaws in the paper or inconsistencies with the claims. The methodology to benchmark the models is sound, repeating the experiments 10 times as was done in Roy et al. 2022. And all small technical innovations have mathematical proofs in the appendix to show they work. In the case of using pseudo-time to train in batches we have included both an analysis of the complexity of the method and supporting literature to support its use over the method of merging irregular time-series.

We also believe even with the reduced claims our paper is still of interest to the TMLR audience. Whilst we have not introduced a signficantly novel methodology, we have still carried out novel evaluation of continuous-time methods on a medical dataset, further supporting the use of Neural CDEs over other continuous models. We have also shown for most cases this outperforms the currently best tested model on this dataset, which we believe is an important finding to encourage further research in this area. We *accept* that this paper does not introduce a ground-breaking new method, however we note this is not part of the TMLR acceptance criteria:

"Crucially, it should *not* be used as a reason to reject work that isn't considered 'significant' or 'impactful' because it isn't achieving a new state-of-the-art on some benchmark. Nor should it form the basis for rejecting work on a method considered not 'novel enough', as novelty of the studied method is not a necessary criteria for acceptance."

And so this should not be a factor when deciding if the paper should be accepted. As we've previously stated, the practical application of Neural CDEs is still in its infancy, and this paper still remains a thorough investigation of how adjusting the methods leads to changes in experimental results. We strongly believe this is not only an interesting read, but also uncovers areas that need further exploration from the community, fitting the second acceptance criteria.

# References

1. Roy, S., Mincu, D., Proleev, L., Rostamzadeh, N., Ghate, C., Harris, N., Chen, C., Schrouff, J., Tomašev, N., Hartsell, F.L. and Heller, K., 2022, April. Disability prediction in multiple sclerosis using performance outcome measures and demographic data. In Conference on Health, Inference, and Learning (pp. 375-396). PMLR.

---

### Author Response · Authors · 2023-08-20
**Polite Request for Further Interaction**

Dear reviewers and action editor,

Given what we believe to be the importance and potential impact of this work, we kindly request the opportunity for more active engagement with the reviewers after our author responses.

Following the initial reviews we have made significant changes in line with the constructive critiques of the paper. We thank the reviewers for the feedback since we believe this has noticeably improved the paper and we are confident it has brought the paper in line with the TMLR acceptance criteria (see our responses for details).

We would appreciate the chance to be able to further improve the paper, but it can only be done with further interactions. What can we do to convince you of the paper's merits?

---

### Decision · Action_Editors · 2023-08-27

**Recommendation:** Accept with minor revision

**Comment:**

The presented work appear to be overall focused on application of CDEs. For an application-focused paper, the empirical results are relatively limited to support the stated contributions.

After author's rebuttal, the reviewers still have major concerns on lacking comparison with more traditional methods for forecasting other than deep learning, and the limited novelty.  The comparing with a single method claiming it as a state of the art and refering to a previous comparison that did not use strong baselines either is not convincing.

As noted that the authors have reduced their claim, and this paper's main claim is that benchmarking continuous models on this MS dataset and comparing to the previous best benchmarked model (TCN), any SOTA claims are removed.  Accordingly, I would like to recommend acceptance.  The authors are strongly encouraged to further revise the paper to address the comments related to empirical results.

**Audience:**

The problem addressed by the work is of high importance and will be interesting for TMLR's audience.

**Claims And Evidence:**

The paper presents benchmarking continuous models on predicting the progression of multiple sclerosis (MS) modeling changes via the NODE. The authors demonstrate that this approach, which intrinsically handles irregular time-series and imputes missing values with physically plausible values (smooth), often outperforms a discrete model in benchmark tests on a public MS dataset.

**Resubmission Of Major Revision:**

The authors may consider submitting a major revision at a later time.

---

> ### Author Response · Authors · 2023-09-09
> **Additional RNN Baseline and our Thanks for the Effort to Review**
>
> To all reviewers and action editor,
>
> As the authors, we'd like to offer our sincere thanks to the reviewers and action editor for their time and effort in reviewing the paper. We are grateful for the feedback which has improved the quality of the paper.
>
> Regarding the requested revision, we agree that the paper would be of higher quality if a further baseline method is included. As such we have newly tested an RNN on the MSOAC dataset, and the results are now included in the camera-ready manuscript. This represents a deep learning model designed to work on sequences not tested in Roy et al. 2022.
>
> The full information and results are now in Appendix A with information about the hyperparameter search and final configurations in Appendix B. Briefly, we use an RNN, where the update of the hidden state is given by $h_{n+1} = f_{\theta}(h_{n}, x_{n+1})$ with $h_{-1} = 0$. We use a ReLU MLP for $f_\theta$, where the hyperparameters are the number of hidden layers and width of the hidden layers (the width of the hidden layer is also the width of the hidden state). We carry out a grid search using 10 fold cross-validation to find the optimal hyperparameters, as is the case in Roy et al. 2022 and our paper. The folds also give us the mean and standard deviations of the test metric. We train using the ADAM optimizer (Kingma et al. 2014) until convergence with learning rate 0.001 and batchsize 1 (as is done for the TCN in Roy et al. 2022). We include the results for TCN, Neural CDE and the newly tested RNN below (these are also in Appendix A):
>
>
> | Prediction Task | Prediction Window | TCN | Neural CDE | RNN |
> | -----------------  | -----------------------|-------|---------------|----|
> |EDSS_Mean       | 0 - 6 mo | **1.264±0.055** | 1.408±0.059 | 1.318±0.049 |
> |   |  6 - 12 mo |1.650±0.067 |**1.627±0.055** |1.681±0.068 |
> |   | 12 - 18 mo | 1.725±0.074 | **1.652±0.066** | 1.801±0.044 |
> |   | 18 - 24 mo |1.666±0.128 |**1.587±0.078** |1.691±0.083|
> | EDSS_Mean > 3 |0 - 6 mo | **0.909±0.014** | 0.908±0.010 | 0.898±0.010 |
> |   | 6 - 12 mo | 0.820±0.027 | **0.852±0.016** |0.818±0.023 |
> |   |12 - 18 mo  |0.768±0.031 | **0.797±0.027** | 0.744±0.036  |
> |   | 18 - 24 mo |0.703±0.038 | **0.742±0.035** | 0.704±0.030 |
> | EDSS_Mean > 5 | 0 - 6 mo | 0.848±0.035 | **0.872±0.021** | 0.856±0.022 |
> |   | 6 - 12 mo | 0.722±0.039 | **0.793±0.039** | 0.738±0.031 |
> |   | 12 - 18 mo | 0.669±0.037 | **0.730±0.031** | 0.560±0.023 |
> |   | 18 - 24 mo | 0.632±0.037 | **0.658±0.056** | 0.472±0.076 |
> | EDSS_Mean  | 0 - 6 mo | **0.782±0.028** | 0.688±0.009 |0.704±0.013 |
> | As Severity Category  | 6 - 12 mo | **0.709±0.044** | 0.672±0.016 | 0.638±0.017 |
> |   | 12 - 18 mo | **0.674±0.037** |0.648±0.022 |0.600±0.026 |
> |   | 18 - 24 mo | **0.632±0.037** | 0.616±0.025 | 0.577±0.018 |
>
> We see that RNN is rarely able to beat TCN, but it does happen. However, RNN is never the best performing model. To quantify this, if we take the rating across the 16 tasks (1 being best performing, 3 being worst), the average ratings for TCN, Neural CDE and RNN are 1.83±0.73, 1.50±0.71 and 2.69±0.46 respectively. This shows that Neural CDE is the best performing, followed by TCN, and then RNN. Crucially it further supports the use of Neural CDEs in this context.
>
> As well as this, there have been some minor aesthetic changes to the paper. We have fixed typos and added the required changes for the TMLR camera-ready manuscript (author information, acknowledgements and author contributions).
>
> Additionally, as we have previously mentioned, there have been other major changes to the paper since the initial submission. We appreciate the reviewers and action editor taking the time to review these, we simply state them here so they are all in one place. For full details see the replies to the reviews. Briefly:
>
> - We removed mention of TCN being state of the art
> - We reduced our claims, instead saying the benchmarking is our primary contribution and the technical solutions we found to do this are secondary contributions
> - Added both theoretical evidence in the form of Big O analysis, and citations to support the use of our batching using pseudo-time over the method of merging and sorting with a binary mask
> - Improved the clarity of writing in particular around the labels in the dataset
>
> We'd just like to say again, we are very grateful to the reviewers and action editor for their time and effort to review the paper. The constructive comments no doubt have improved the quality of the paper.
>
> # References
>
> 1. Roy, S., Mincu, D., Proleev, L., Rostamzadeh, N., Ghate, C., Harris, N., Chen, C., Schrouff, J., Tomašev, N., Hartsell, F.L. and Heller, K., 2022, April. Disability prediction in multiple sclerosis using performance outcome measures and demographic data. In Conference on Health, Inference, and Learning (pp. 375-396). PMLR.
> 2. Kingma, D.P. and Ba, J., 2014. Adam: A method for stochastic optimization. arXiv preprint arXiv:1412.6980.